# GrokFormer: Graph Fourier Kolmogorov-Arnold Transformers

Guoguo Ai [1] [*]   Guansong Pang [2]   Hezhe Qiao [2]   Yuan Gao [1]   Hui Yan [1] [†]

## Abstract

Graph Transformers (GTs) have demonstrated remarkable performance in graph representation learning over popular graph neural networks (GNNs). However, self–attention, the core module of GTs, preserves only low-frequency signals in graph features, leading to ineffectiveness in capturing other important signals like high-frequency ones. Some recent GT models help alleviate this issue, but their flexibility and expressiveness are still limited since the filters they learn are fixed on predefined graph spectrum or spectral order. To tackle this challenge, we propose a Graph Fourier Kolmogorov-Arnold Transformer (**GrokFormer**), a novel GT model that learns highly expressive spectral filters with adaptive graph spectrum and spectral order through a Fourier series modeling over learnable activation functions. We demonstrate theoretically and empirically that the proposed GrokFormer filter offers better expressiveness than other spectral methods. Comprehensive experiments on 11 real-world node classification datasets across various domains, scales, and graph properties, as well as 5 graph classification datasets, show that GrokFormer outperforms state-of-the-art GTs and GNNs. Our code is available at https://github.com/GGA23/GrokFormer.

## 1. Introduction

Graph neural networks (GNNs), which jointly encode graph structures and node features, have been emerging as an effective generic tool for graph-structured learning problems (Yi et al., 2023; Qiao et al., 2024). Despite their effective-

---

[*]The work was partly done when Guoguo Ai visited Singapore Management University. [1]School of Computer Science and Engineering, Nanjing University of Science and Technology, Nanjing, China [2]School of Computing and Information Systems, Singapore Management University, Singapore. Correspondence to: Hui Yan <yanhui@njust.edu.cn>.

*Proceedings of the $42^{nd}$ International Conference on Machine Learning*, Vancouver, Canada. PMLR 267, 2025. Copyright 2025 by the author(s).

ness, popular GNNs are often limited by issues like over-smoothing (Oono & Suzuki, 2020) and over-squashing (Topping et al., 2022). On the other hand, graph Transformers (GTs) use a Transformer-based architecture (Vaswani et al., 2017) to learn graph representations. Due to its strong capability in capturing long-range dependencies among graph nodes, it offers a potential solution to address the issues in popular GNNs.

One key ingredient to successful GTs is to effectively integrate topological structure information into the Transformer network. This may be achieved by various position encoding methods such as Laplacian vectors and random walks (Zhang et al., 2020; Dwivedi & Bresson, 2020; Kreuzer et al., 2021; Kim et al., 2022; Wu et al., 2021). Other information such as graph distances and path embeddings can also be incorporated into GTs through their attention mechanism to improve the performance (Maziarka et al., 2020; Ying et al., 2021; Chen et al., 2022; Choromanski et al., 2022; Wu et al., 2024). However, despite the remarkable success in graph representation learning, their performance can be severely limited by the inherent low-pass nature of the self-attention module, since it only preserves low-frequency signals that highlight similarity between nodes (Bastos et al., 2022; Wang et al., 2022; Shi et al., 2022). This prevents GTs from capturing other important frequency signals, *e.g.*, high-frequency signals that highlight difference between nodes, which can be crucial for learning complex relationships of nodes in diverse graphs.

To address this issue, inspired by polynomial GNNs (*e.g.*, ChebyNet (Defferrard et al., 2016), GPRGNN(Chien et al., 2021), BernNet(He et al., 2021),JacobiConv (Wang & Zhang, 2022)) and A2GCN (Ai et al., 2024), some recent methods are dedicated to capturing various frequency signals by order-$K$ polynomial approximation. For example, FeTA (Bastos et al., 2022) and PolyFormer (Ma et al., 2024) learn the coefficients for order-$K$ polynomial bases (*e.g.*, Chebyshev, Monomial, or Bernstein basis) through the self-attention mechanism. However, these polynomial filters are typically approximated via $K$ predefined bases with specific frequency responses as illustrated in Figure 1(a), and thus, they have a receptive field of size $K$ in information passing, leading to a locality modeling and limited flexibility and expressivity. Consequently, they are often unable to fit complicated graph filters well, as shown by the failure of the

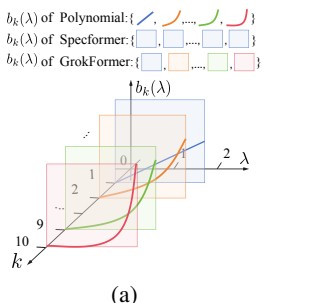 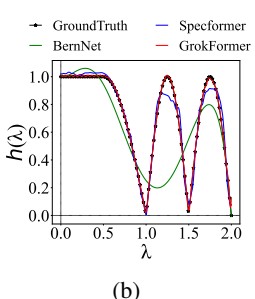

(a)  (b)

Figure 1: (a) The frequency response range of $K$ filter bases $\{b_1(\lambda), b_2(\lambda), \cdots, b_{k=K}(\lambda)\}, k \in [1, K]$ for GrokFormer, Specformer, and polynomial filters at the spectrum $\lambda$ w.r.t. spectral order $k$, where colors represent the varying frequency components of spectrum at different orders. Polynomial filters typically have fixed bases, *e.g.*, $\lambda, \lambda^2, \cdots, \lambda^K$, corresponding to the $K$ filter curves that capture the specific curvilinear frequencies, whereas Specformer adaptively learns the filter bases at the first-order spectrum, enabling it to capture arbitrary frequency responses in the spectrum plane of $k = 1$. In contrast, our GrokFormer filter bases are capable of capturing arbitrary frequency responses across $K$ different spectral planes. (b) Low-comb filter (ground truth) and the approximated filters generated by the filters of GorkFormer and Specformer, and the Bernstein polynomial filter in BernNet.

prevalent Bernstein polynomial in fitting a low-comb filter in Figure 1(b). To effectively encode such spectrum information and achieve a more global graph modeling, Specformer (Bo et al., 2023) performs self-attention over the $N$ eigenvalues after positional encoding to build learnable filter bases. As shown in Figure 1(a), the frequency response of its filter bases can be arbitrary on the first-order graph Laplacian spectrum. Although it shows impressive effectiveness, its filter learning has a computational complexity of $O(N^2)$, making it difficult to simultaneously capture higher-order spectral information, and thus misses some important frequency components embedded in higher-order spectrum. Therefore, the learning capacity of the Specformer filter is limited to the specific first-order spectrum and thus it struggles to fit the complicated low-comb filter in Figure 1(b). Accordingly, the key question we ask here is: *can we have a GT that efficiently and flexibly extracts rich frequency signals across the multi-order spectrum of the graph Laplacian?*

To answer this question, we propose a novel GT model, called Graph Fourier Kolmogorov-Arnold Transformers (**GrokFormer**), which provides an efficient approach for learning order- and spectrum-adaptive graph filter for GTs. In particular, motivated by Kolmogorov-Arnold Networks (KANs) (Liu et al., 2024), GrokFormer leverages learn-

Table 1: Our proposed filter vs. existing spectral filters.

| | Models | Order-adaptive | Spectrum-adaptive |
|---|---|---|---|
| GNNs | ChebyNet, GPRGNN, BernNet, JacobiConv | ✓ | ✗ |
| GTs | FeTA, PolyFormer | ✓ | ✗ |
| | Specformer | ✗ | ✓ |
| | GrokFormer (Ours) | ✓ | ✓ |

able activation functions modeled as Fourier series over $K$-order graph Laplacian spectrum, producing $K$ adaptive filter bases. As shown in Figure 1(a), these bases can capture any frequency response across the spectrum of both low and high orders (*i.e.*, from 1st to $K$th orders). Furthermore, we devise learnable order coefficients to assign varying importance to the $K$ filter bases, enabling an adaptive adjustment in fitting the graph spectral order. The resulting filter in GrokFormer is adaptive/learnable in both graph spectrum and spectral order, having better adaptivity than existing popular filters, as shown in Table 1. In doing so, GrokFormer filter can more flexibly capture a broader range frequency responses, offering significantly better expressiveness than other filters (see Figure 1(b)). The main contributions are as follows:

- We propose a novel GT model, GrokFormer, that can effectively capture a wide range of frequency signals in an order- and spectrum-adaptive manner. To the best of our knowledge, this is the first GT model that has a learnable filter in both graph spectrum and spectral order.

- We further introduce a graph filter learning approach, namely Graph Fourier KAN, that leverages learnable activation functions modeled as Fourier series to learn a set of spectral filter bases. The learned filter enables GrokFormer to model diverse frequency signals from a broad graph spectrum of both low and high order.

- We theoretically show that the GrokFormer filter offers better learning ability than state-of-the-art (SOTA) competing filters, and empirically demonstrate the superiority of GrokFormer over SOTA GNNs and GTs on real-world node- and graph-level datasets.

## 2. Related Work

**Graph Neural Networks.** Existing GNNs are mainly divided into two main streams: spatial-based and spectral-based methods. Spatial-based GNNs, like GCN (Kipf & Welling, 2017), SGC (Wu et al., 2019) and GAT (Veličković et al., 2018), update node representations by aggregating information from neighbors. By stacking multiple layers, they may learn long-range dependencies but suffer from over-smoothing and over-squashing. Some improved spatial methods, such as H2GCN (Zhu et al., 2020), HopGNN

(Chen et al., 2023c) and SHGCN (Yan et al., 2025) propose to combine first-hop and multi-hop neighborhood representations. Other studies (Xu et al., 2019; Dong et al., 2021) point out from a spectral perspective that GCN only considers the first-order Chebyshev polynomial, which acts as a low-pass filter. Subsequently, various spectral-based GNNs have been proposed, such as, GPRGNN (Chien et al., 2021), BernNet (He et al., 2021) and JacobiConv (Wang & Zhang, 2022) learn arbitrary graph spectral filters by order-$K$ polynomial approximation. HiGCN (Huang et al., 2024) uses Flower-Petals Laplacians in simplicial complexes to learn polynomial filters across varying topological scales. However, the information passing in these polynomial models is local, and their filters with fixed bases have limited learning ability.

**Graph Transformers.** Compared to GNNs, the attention weights in Transformers can be viewed as a weighted adjacency matrix of a fully connected graph, capturing long-range dependencies. Some GTs combine both and are popular in graph representation learning, such as Graphormer (Ying et al., 2021), GraphGPS (Rampášek et al., 2022), GRIT (Ma et al., 2023), SAT (Chen et al., 2022), Node-Former (Wu et al., 2022), NAGphormer (Chen et al., 2023a), GCFormer (Chen et al., 2024b), and SGFormer (Wu et al., 2024) are proposed by incorporating various graph structural information into the Transformer architecture. However, these GTs are limited by the inherent low-pass nature of the self-attention mechanism (Bastos et al., 2022). Advanced GTs have increasingly focused on capturing various frequency signals to tackle the issue. SignGT (Chen et al., 2023b) designs a signed self-attention mechanism to capture low- and high-frequency signals. FeTA (Bastos et al., 2022) and PolyFormer (Ma et al., 2024) extract various frequency information via polynomial approximation like polynomial GNNs. Specformer (Bo et al., 2023) develops learnable filter bases, offering greater spectral expressiveness compared to polynomials with fixed bases. However, such spectral filters still struggle to achieve the desired frequency response due to their limited focus on the specific first-order spectrum.

## 3. Preliminaries

### 3.1. Notations

An attributed graph is represented as $\mathcal{G} = (\mathcal{V}, \mathcal{E}, \mathbf{X})$, where $\mathcal{V}$ denotes the node set with $v_i \in \mathcal{V}$ and $|\mathcal{V}| = N$, $\mathcal{E}$ denotes the edge set, and $\mathbf{X} \in \mathbb{R}^{N \times F}$ is a set of node attributes. Each $v_i$ has a $F$-dimensional feature representation $x_i$. The topological structure of $\mathcal{G}$ is represented by an adjacency matrix $\mathbf{A} = [a_{ij}] \in \mathbb{R}^{N \times N}$, $a_{ij} = a_{ji} = 1$ if $(v_i, v_j) \in \mathcal{E}$, and $a_{ij} = a_{ji} = 0$ otherwise. $\mathbf{D} \in \mathbb{R}^{N \times N}$ denotes a diagonal degree matrix with $\mathbf{d}_{ii} = \sum_j a_{ij}$. The normalized Laplacian matrix $\mathbf{L}$ is defined by $\mathbf{L} = \mathbf{I}_N - \mathbf{D}^{-\frac{1}{2}} \mathbf{A} \mathbf{D}^{-\frac{1}{2}}$, where $\mathbf{I}_N \in \mathbb{R}^{N \times N}$ denotes an identity matrix.

### 3.2. Graph Filter

$\mathbf{L} = \mathbf{U}\Lambda\mathbf{U}^\top$ denotes the spectral decomposition of a Laplacian matrix, where $\mathbf{U} = (u_1, u_2, \ldots, u_N)$ is a complete set of orthonormal eigenvectors, also known as graph Fourier modes, and $\Lambda = \text{diag}\left(\{\lambda_i\}_{i=1}^N\right)$ is a diagonal matrix of the eigenvalues of $\mathbf{L}$. The Fourier transform of a graph signal $\boldsymbol{x} \in \mathbb{R}^{N \times 1}$ is written as $\hat{\boldsymbol{x}} = \mathbf{U}^\top \boldsymbol{x}$. The inverse transform is $\boldsymbol{x} = \mathbf{U}\hat{\boldsymbol{x}}$ (Shuman et al., 2013). Per convolution theorem, the convolution of the graph signal $\boldsymbol{x}$ with a spectral filter $G$ having its frequency response as $h$ can be obtained by:

$$\boldsymbol{x} * G = \mathbf{U}h(\Lambda)\mathbf{U}^\top\boldsymbol{x} = \mathbf{U}diag[h(\lambda_1), \cdots, h(\lambda_N)]\mathbf{U}^\top\boldsymbol{x}, \quad (1)$$

where $h(\Lambda)$ applies $h$ element-wisely to the diagonal entries of $\Lambda$, *i.e.*, $[h(\Lambda)]_{ii} = h(\lambda_i)$. A powerful spectral filter can exploit useful frequency components in graphs.

### 3.3. Self-Attention

Multi-head self-attention is a key module of Transformers, having strong ability to capture interactions between any pair of input instances, *e.g.*, graph nodes in GTs. Let $\mathbf{X}$ denote the input of self-attention, and for simplicity of illustration, we consider the single-head self-attention in the equation below. It first projects $\mathbf{X}$ into three subspaces query $\mathbf{Q}$, key $\mathbf{K}$, and value $\mathbf{V}$ through three projection matrices $\mathbf{W}^Q, \mathbf{W}^K, \mathbf{W}^V$. The self-attention is then calculated as:

$$Attetion(\mathbf{Q}, \mathbf{K}, \mathbf{V}) = softmax(\frac{\mathbf{Q}\mathbf{K}^\top}{\sqrt{d}})\mathbf{V}, \quad (2)$$

where $d$ is query dimension, $\mathbf{Q} = \mathbf{X}\mathbf{W}^Q$, $\mathbf{K} = \mathbf{X}\mathbf{W}^K$, and $\mathbf{V} = \mathbf{X}\mathbf{W}^V$.

### 3.4. Kolmogorov-Arnold Network

KAN is grounded in the Kolmogorov-Arnold representation theorem (Kolmogorov, 1957; Ismayilova & Ismailov, 2024), which states that for a function $f$:

$$f(x_1, \ldots, x_n) = \sum_{q=1}^{2n+1} \Phi_q(\sum_{p=1}^{n} \phi_{q,p}(x_p)), \quad (3)$$

where $\phi_{q,p}$ is trainable activation function, and $\Phi_q$: $[0,1] \to \mathbb{R}$ and $\phi_{q,p} : \mathbb{R} \to \mathbb{R}$ are univariate functions that map each input variable $x_p$. It create an arbitrary function at each hidden neuron by overlaying multiple nonlinear functions onto the input features. For a single-layer KAN $\Phi$ with an input dimension of $n_{in}$ and the output dimension of $n_{out}$:

$$x_j^{\text{out}} = \sum_{i=1}^{n_{\text{in}}} \phi_{i,j}\left(x_i^{\text{in}}\right), \quad (4)$$

where $x_i$ denotes the $i$-th dimension of $x$, and $\phi_{i,j}$ represents a learnable nonlinear function, often parameterized as a linear combination of B-splines (Liu et al., 2024).

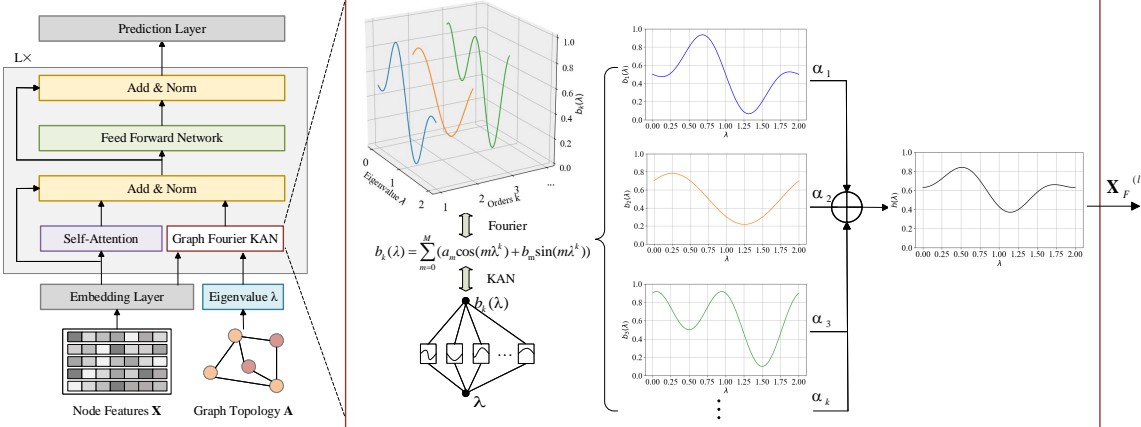

Figure 2: Overview of GrokFormer. In addition to the use of self-attention to capture global information in the spatial domain, a novel Graph Fourier KAN is proposed in GrokFormer the achieve global graph modeling in the spectral domain. This design enables a strong adaptability in both spectral order and graph spectrum, offering superior expressive power in capturing diverse graph frequency signals. GrokFormer synthesizes the spatial and spectral representations by a standard summation and normalization layer, followed by a Feed-Forward Network (FFN) layer for prediction.

# 4. Methodology

GrokFormer is a novel GT framework empowered by a Graph Fourier Kolmogorov-Arnold Network (KAN)-based spectral graph convolutional filter, as shown in Figure 2. Graph Fourier KAN in GrokFormer is devised in a way that can adaptively learn diverse frequency signals from a wide range of spectral order and graph spectrum, going beyond the self-attention mechanism in GTs.

## 4.1. The Proposed GrokFormer Filter

**The Formulation.** To capture various frequencies in a flexible and efficient manner, we design a novel spectral graph convolution module, named Graph Fourier KAN. Motivated by KAN, which use learnable functions parameterized as splines instead of traditional weight parameters to achieve parameter-efficient learning, we devise learnable functions to learn eigenvalue-specific filter functions over the order-$K$ spectrum of the graph Laplacian, thereby improving the expressiveness of the filters. However, the spline in KAN is piecewise and difficult to train (Xu et al., 2024), which does not meet our goal to develop an efficient filter learning method. To address this issue, we turn to finding multiple relatively simple nonlinear functions. To this end, we propose a novel approach that leverages Fourier series representation to parameterize each learnable function. The specific filter function can be accordingly defined as follows:

$$\phi_h(\lambda) = \sum_{k=1}^{K} \sum_{m=0}^{M} \left( \cos\left(m\lambda^k\right) \cdot a_{km} + \sin\left(m\lambda^k\right) \cdot b_{km} \right),$$
(5)

where $K$ is the highest order the filter can model, $M$ represents the number of frequency components (or grid size),

and both of which are hyperparameters; $a_{km}$ and $b_{km}$ are trainable Fourier coefficients.

Compared to existing popular graph filters, $\phi_h(\lambda)$ has the following three advantages. (i) Effectiveness: The orthogonality of polynomial bases is a nice property in learning filters (Wang & Zhang, 2022; Bo et al., 2023). Sine and cosine in the Fourier series are orthogonal, our graph Fourier KAN inherits this property, enabling effective learning of graph filters. Also, many sine and cosine terms with different frequency components can well support the modeling of rich frequency information in our filter. (ii) Convergence guarantee: In approximation theory (Pinkus, 2000), the fact that the Fourier series attains the best convergence rate for function approximation supports the fast convergence for our method. (iii) Global graph modeling: The filter function can effectively attend to all eigenvalues, allowing the learned graph Laplacian to construct a fully connected graph that captures global information (Bo et al., 2023).

**Order and Spectrum Adaptability.** The filter is aimed to adaptively consider a variety of graph Laplacian spectrum from the 1st order up to the $K$th order ($K$-order graph spectrum for short). To adaptively capture diverse frequency patterns across different orders, we rewrite Eq. (5) and define a set of learnable bases at a specific order $k$ as follows:

$$b_k(\lambda) = \sum_{m=0}^{M} \left( \cos\left(m\lambda^k\right) \cdot a_m + \sin\left(m\lambda^k\right) \cdot b_m \right), \quad (6)$$

where $k = [1, 2, \cdots, K]$. Subsequently, we introduce a learnable order coefficient $\alpha_k$ to adaptively synthesize the

filter bases from a wide range of orders as follows:

$$h(\lambda) = \sum_{k=1}^{K} \alpha_k b_k(\lambda) \qquad (7)$$

Therefore, the corresponding spectral graph convolution in GrokFormer is defined as follows,

$$\mathbf{X}_F^{(l)} = \mathbf{U} diag(h(\lambda)) \mathbf{U}^\top \mathbf{X}^{(l-1)}, \qquad (8)$$

where $diag(\cdot)$ creates a diagonal matrix, $\mathbf{X}^{(0)} = f_\theta(\mathbf{X})$, and $f_\theta$ is a two-layer MLP (embedding layer).

**Theoretical Analysis.** Below we show theoretically that our proposed filter can flexibly fit graph patterns of any spectral order $K$ and graph spectrum.

**Proposition 4.1.** *Our graph filter $h(\lambda)$ is learnable in both spectral order and graph spectrum:*

$$h(\lambda) = \sum_{k=1}^{K} \alpha_k \sum_{m=0}^{M} \left( \cos\left(m\lambda^k\right) \cdot a_{km} + \sin\left(m\lambda^k\right) \cdot b_{km} \right),$$
$$(9)$$

*where the spectral order $k$ is adaptively determined by co-efficient $\alpha_k$ while the spectrum $\lambda$ at the specific order $k$ is adaptively determined by coefficients $a_{km}$ and $b_{km}$.*

Due to this adaptability, existing advanced filters are special cases of our GrokFormer filter, showing its better universality and flexibility in graph pattern modeling.

**Proposition 4.2.** *Existing polynomial filters that can be formulated as $h(\lambda) = \sum_{k=0}^{K} \alpha_k \lambda^k$ are a simplified variant of our graph filter.*

**Proposition 4.3.** *The graph filter in Specformer is a simplified variant of our graph filter.*

We further show below in Proposition 4.4 that GrokFormer filter has strong expressiveness and can learn permutation-equivariant node representations.

**Proposition 4.4.** *Our filter $h(\lambda)$ can approximate any continuous function and constructs a permutation-equivariant spectral graph convolution.*

All proofs are provided in Appendix C.

### 4.2. Network Architecture of GrokFormer

GrokFormer is built upon the original implementation of a classic Transformer encoder. Specifically, we apply layer normalization (LN) on the representations before feeding them into other sub-layers, *i.e.*, the multi-head self-attention (MHA) and the feed-forward blocks (FFN). Here, we use an efficient MHA (EMHA) that switches the order from $(\mathbf{QK}^\top)\mathbf{V}$ to $\mathbf{Q}(\mathbf{K}^\top\mathbf{V})$ of Eq. (2) (Shen et al., 2021), which helps reduce the complexity without affecting performance. We synthesize the representations from both

the EMHA module and the proposed Graph Fourier KAN module through summation to generate informative node representations. We formally characterize the GrokFormer layer as follows:

$$\mathbf{X}^{'(l)} = EMHA(LN(\mathbf{X}^{(l-1)})) + \mathbf{X}^{(l-1)} + \mathbf{X}_F^{(l)},$$
$$\mathbf{X}^{(l)} = FFN(LN(\mathbf{X}^{'(l)})) + \mathbf{X}^{'(l)}. \qquad (10)$$

In the final layer of GrokFormer, we calculate the prediction scores of the nodes from class $c$. This score is given by:

$$\hat{\mathbf{Y}} = softmax(\mathbf{X}^{(L)}), \qquad (11)$$

where $\mathbf{X}^{(L)}$ is the output of the final layer, and $\hat{\mathbf{Y}}$ is the predicted class label.

Then, GrokFormer can be trained by minimizing the cross entropy between the predicted and the ground-truth labels:

$$\mathcal{L}_{\text{ce}} = -\sum_{i \in \mathcal{V}_L} \sum_{c=1}^{C} \mathcal{Y}_{ic} \ln \hat{\mathbf{Y}}_{ic}, \qquad (12)$$

where $C$ is the number of classes, $\mathcal{Y}$ is the real labels, and $\mathcal{V}_L$ is the training set.

### 4.3. Complexity and Scalability Analysis

**Complexity.** Firstly, like previous methods (Bo et al., 2023; 2024), GrokFormer also needs spectral decomposition, which is done offline in the preprocessing step and has the complexity of $O(N^3)$. Secondly, GrokFormer's forward process involves an embedding layer with the complexity of $O(Nd^2)$, efficient self-attention with complexity of $O(Nd^2)$, filter base learning with complexity of $O(KNM)$, and graph convolution with complexity of $O(N^2d)$. Note that explicitly constructing the spectral filter matrix $\mathbf{U}diag(h(\lambda))\mathbf{U}^\top$ in Eq. (8) incurs a high computational cost of $O(N^3)$. To address this, we leverage matrix associativity and compute $\mathbf{U}^\top\mathbf{X}$ first, which reduce the complexity to $O(N^2d)$. As a result, the overall forward pass complexity of GrokFormer is $O(Nd(N + 2d) + KNM)$.

**Scalability.** In large graphs, GrokFormer can use Sparse Generalized Eigenvalue (SGE) algorithms, as outlined in earlier studies (Cai et al., 2021; Bo et al., 2023; 2024), to compute $q$ eigenvalues and corresponding eigenvectors, in which case the decomposition complexity and the forward complexity will reduce to $O(N^2q)$ ($q \ll N$) and $O(2Nd^2 + KqM + Nqd)$, respectively. Empirical results for computational cost can be found in Section 5.5.

## 5. Experiments

In this section, we conduct comprehensive experiments on both synthetic and real-world datasets to verify the effectiveness of our GrokFormer. More experiments can be seen in Appendix B.

Table 2: Node classification results on five homophilic and five heterophilic datasets: mean accuracy (%) ± std. The best results are in bold, while the second-best ones are underlined. 'OOM' means out of memory

| | Homophilic Datasets | | | | | | Heterophilic Datasets | | | | |
|---|---|---|---|---|---|---|---|---|---|---|---|
| | Cora | Citeseer | Pubmed | Photo | WikiCS | Physics | Penn94 | Chameleon | Squirrel | Actor | Texas |
| Spatial-based GNNs | | | | | | | | | | | |
| GCN | $87.14_{\pm1.01}$ | $79.86_{\pm0.67}$ | $86.74_{\pm0.27}$ | $88.26_{\pm0.73}$ | $82.32_{\pm0.69}$ | $97.74_{\pm0.35}$ | $82.47_{\pm0.27}$ | $59.61_{\pm2.21}$ | $46.78_{\pm0.87}$ | $33.23_{\pm1.16}$ | $77.38_{\pm3.28}$ |
| GAT | $88.03_{\pm0.79}$ | $80.52_{\pm0.71}$ | $87.04_{\pm0.24}$ | $90.94_{\pm0.68}$ | $83.22_{\pm0.78}$ | $97.82_{\pm0.28}$ | $81.53_{\pm0.55}$ | $63.13_{\pm1.93}$ | $44.49_{\pm0.88}$ | $33.93_{\pm2.47}$ | $80.82_{\pm2.13}$ |
| H2GCN | $87.96_{\pm0.37}$ | $80.90_{\pm1.21}$ | $89.18_{\pm0.28}$ | $95.45_{\pm0.67}$ | $83.45_{\pm0.26}$ | $97.19_{\pm0.13}$ | $81.31_{\pm0.60}$ | $61.20_{\pm4.28}$ | $39.53_{\pm0.88}$ | $35.86_{\pm2.58}$ | $91.89_{\pm3.93}$ |
| HopGNN | $88.68_{\pm1.06}$ | $80.38_{\pm0.68}$ | $89.15_{\pm0.35}$ | $94.49_{\pm0.33}$ | $84.73_{\pm0.59}$ | $97.86_{\pm0.16}$ | OOM | $65.25_{\pm3.49}$ | $57.83_{\pm2.11}$ | $39.33_{\pm2.79}$ | $89.15_{\pm4.04}$ |
| Spectral-based GNNs | | | | | | | | | | | |
| ChebyNet | $86.67_{\pm0.82}$ | $79.11_{\pm0.75}$ | $87.95_{\pm0.28}$ | $93.77_{\pm0.32}$ | $82.95_{\pm0.45}$ | $97.25_{\pm0.78}$ | $81.09_{\pm0.33}$ | $59.28_{\pm1.25}$ | $40.55_{\pm0.42}$ | $37.61_{\pm0.89}$ | $86.22_{\pm2.45}$ |
| GPRGNN | $88.57_{\pm0.69}$ | $80.12_{\pm0.83}$ | $88.46_{\pm0.33}$ | $93.85_{\pm0.28}$ | $82.58_{\pm0.89}$ | $97.25_{\pm0.13}$ | $81.38_{\pm0.16}$ | $67.28_{\pm1.09}$ | $50.15_{\pm1.92}$ | $39.92_{\pm0.67}$ | $92.95_{\pm1.31}$ |
| BernNet | $88.52_{\pm0.95}$ | $80.09_{\pm0.79}$ | $88.48_{\pm0.41}$ | $93.63_{\pm0.35}$ | $83.56_{\pm0.61}$ | $97.36_{\pm0.30}$ | $82.47_{\pm0.21}$ | $68.29_{\pm1.58}$ | $51.35_{\pm0.73}$ | $41.79_{\pm1.01}$ | $93.12_{\pm0.65}$ |
| JacobiConv | $88.98_{\pm0.46}$ | $80.78_{\pm0.79}$ | $89.62_{\pm0.41}$ | $95.43_{\pm0.23}$ | $84.13_{\pm0.49}$ | $97.56_{\pm0.23}$ | $83.35_{\pm0.11}$ | $74.20_{\pm1.03}$ | $57.38_{\pm1.25}$ | $41.17_{\pm0.64}$ | $\underline{93.44}_{\pm2.13}$ |
| HiGCN | $\underline{89.23}_{\pm0.23}$ | $81.12_{\pm0.28}$ | $89.95_{\pm0.13}$ | $95.33_{\pm0.37}$ | $83.14_{\pm0.78}$ | $97.65_{\pm0.35}$ | OOM | $68.47_{\pm0.45}$ | $51.86_{\pm0.42}$ | $41.81_{\pm0.52}$ | $92.15_{\pm0.73}$ |
| Graph Transformers | | | | | | | | | | | |
| Transformer | $71.83_{\pm1.68}$ | $70.55_{\pm1.20}$ | $86.66_{\pm0.50}$ | $89.58_{\pm1.05}$ | $77.36_{\pm1.25}$ | OOM | OOM | $45.21_{\pm2.01}$ | $33.17_{\pm1.32}$ | $39.95_{\pm0.64}$ | $88.75_{\pm6.30}$ |
| GraphGPS | $83.42_{\pm1.22}$ | $75.87_{\pm0.71}$ | $86.62_{\pm0.53}$ | $94.35_{\pm0.25}$ | $79.26_{\pm0.57}$ | $97.60_{\pm0.05}$ | OOM | $46.07_{\pm1.51}$ | $34.14_{\pm0.73}$ | $37.68_{\pm0.94}$ | $83.71_{\pm5.85}$ |
| NodeFormer | $87.32_{\pm0.92}$ | $79.56_{\pm1.10}$ | $89.24_{\pm0.23}$ | $95.27_{\pm0.22}$ | $81.03_{\pm0.94}$ | $96.45_{\pm0.28}$ | $69.66_{\pm0.83}$ | $56.34_{\pm1.11}$ | $43.42_{\pm1.62}$ | $34.62_{\pm1.82}$ | $84.63_{\pm3.47}$ |
| SGFormer | $87.87_{\pm2.67}$ | $79.62_{\pm1.63}$ | $89.07_{\pm0.14}$ | $94.34_{\pm0.23}$ | $82.71_{\pm0.56}$ | $97.96_{\pm0.81}$ | $76.65_{\pm0.49}$ | $61.44_{\pm1.37}$ | $45.82_{\pm2.17}$ | $41.69_{\pm0.63}$ | $92.46_{\pm1.48}$ |
| NAGphormer | $88.15_{\pm1.35}$ | $80.12_{\pm1.24}$ | $89.70_{\pm0.19}$ | $\underline{95.49}_{\pm0.11}$ | $83.41_{\pm0.34}$ | $97.85_{\pm0.26}$ | $73.98_{\pm0.53}$ | $54.92_{\pm1.11}$ | $48.55_{\pm2.56}$ | $40.08_{\pm1.50}$ | $91.80_{\pm1.85}$ |
| Specformer | $88.57_{\pm1.01}$ | $\underline{81.49}_{\pm0.94}$ | $90.61_{\pm0.23}$ | $95.48_{\pm0.32}$ | $85.15_{\pm0.63}$ | $97.75_{\pm0.53}$ | $\mathbf{84.32}_{\pm0.32}$ | $\underline{74.72}_{\pm1.29}$ | $\underline{64.64}_{\pm0.81}$ | $41.93_{\pm1.04}$ | $88.23_{\pm0.38}$ |
| PolyFormer | $87.67_{\pm1.28}$ | $81.80_{\pm0.76}$ | $\underline{90.68}_{\pm0.31}$ | $94.08_{\pm1.37}$ | $83.62_{\pm0.17}$ | $\underline{98.08}_{\pm0.27}$ | $79.27_{\pm0.26}$ | $60.17_{\pm1.39}$ | $44.98_{\pm3.03}$ | $41.51_{\pm0.71}$ | $89.02_{\pm5.44}$ |
| GrokFormer | $\mathbf{89.57}_{\pm1.43}$ | $\mathbf{81.92}_{\pm1.25}$ | $\mathbf{91.39}_{\pm0.51}$ | $\mathbf{95.52}_{\pm0.52}$ | $\mathbf{85.57}_{\pm0.65}$ | $\mathbf{98.31}_{\pm0.18}$ | $\underline{83.59}_{\pm0.26}$ | $\mathbf{75.58}_{\pm1.73}$ | $\mathbf{65.12}_{\pm1.59}$ | $\mathbf{42.98}_{\pm1.48}$ | $\mathbf{94.59}_{\pm2.08}$ |

## 5.1. Performance for Node Classification

**Dataset Description.** We conduct node classification experiments on 11 widely used datasets in previous graph spectral models (Bo et al., 2023; He et al., 2021; Deng et al., 2024), including six homophilic datasets, *i.e.*, Cora, Citeseer, Pubmed, the Amazon co-purchase graph Photo (He et al., 2021), an extracted subset of Wikipedia's Computer Science articles–WikiCS (Dwivedi et al., 2023), and a co-authorship network Physics (Shchur et al., 2018; Chen et al., 2024a). We also evaluate on five heterophilic datasets, *i.e.*, Wikipedia graphs Chameleon and Squirrel, the Actor co-occurrence graph (Pei et al., 2020), webpage graphs Texas from WebKB, and Penn94, a large-scale friendship network from the Facebook 100 (Lim et al., 2021). A more detailed description can be found in Appendix A.1.

**Baselines and Settings.** We compare GrokFormer with sixteen competitive baselines, including four spatial-based GNNs, five spectral-based GNNs, and seven GTs. Note that PolyFormer has multiple variants, and we use the PolyFormer(Cheb) as the baseline. Following the previous works (He et al., 2021; Huang et al., 2024; Bo et al., 2023), we randomly split the node set into train/validation/test set with ratio 60%/20%/20%, and generate 10 random splits to evaluate all models on the same splits. We report the average classification accuracy and standard deviation for each model. For polynomial GNNs, we set the order of polynomials $K$ = 10, consistent with their original setting. For the baseline models, we adopt the hyperparameters provided by the authors. In the large-scale datasets Physics and Penn94, we implement truncated spectral decomposition for both Grok-Former and Specformer to enhance scalability, selecting the 3,000 eigenvectors associated with the smallest (low-

Table 3: Graph classification results.

| | PROTEINS | MUTAG | PTC-MR | IMDB-B | IMDB-M |
|---|---|---|---|---|---|
| Kernel methods | | | | | |
| GK | $71.4_{\pm0.3}$ | $81.7_{\pm2.1}$ | $55.3_{\pm1.4}$ | $65.9_{\pm1.0}$ | $43.9_{\pm0.4}$ |
| WL kernel | $75.0_{\pm3.1}$ | $90.4_{\pm5.7}$ | $59.9_{\pm4.3}$ | $73.8_{\pm3.9}$ | $50.9_{\pm3.8}$ |
| DGK | $71.7_{\pm0.5}$ | $82.7_{\pm1.4}$ | $57.3_{\pm1.1}$ | $67.0_{\pm0.6}$ | $44.6_{\pm0.5}$ |
| GNN methods | | | | | |
| DGCNN | $75.5_{\pm0.9}$ | $85.8_{\pm1.8}$ | $58.6_{\pm2.5}$ | $70.0_{\pm10.9}$ | $47.8_{\pm10.9}$ |
| GCN | $75.2_{\pm2.8}$ | $85.1_{\pm5.8}$ | $63.1_{\pm4.3}$ | $73.8_{\pm3.4}$ | $55.2_{\pm0.3}$ |
| GIN | $76.2_{\pm2.8}$ | $89.4_{\pm5.6}$ | $64.6_{\pm7.0}$ | $75.1_{\pm5.1}$ | $52.3_{\pm2.8}$ |
| GDN | $\mathbf{81.3}_{\pm3.1}$ | $\underline{97.4}_{\pm2.7}$ | $75.6_{\pm7.6}$ | $79.3_{\pm3.3}$ | $55.2_{\pm4.3}$ |
| HiGCN | $77.0_{\pm4.2}$ | $91.3_{\pm6.4}$ | $66.2_{\pm6.9}$ | $76.2_{\pm5.1}$ | $52.7_{\pm3.5}$ |
| Graph Transformers | | | | | |
| Transformer | $66.3_{\pm8.4}$ | $81.9_{\pm9.7}$ | $57.3_{\pm7.0}$ | $71.1_{\pm3.8}$ | $45.8_{\pm3.8}$ |
| Graphormer | $68.5_{\pm2.3}$ | $82.5_{\pm3.8}$ | $59.2_{\pm4.6}$ | $73.5_{\pm3.8}$ | $48.9_{\pm2.3}$ |
| SGFormer | $74.6_{\pm3.0}$ | $88.6_{\pm6.3}$ | $65.2_{\pm4.2}$ | $74.7_{\pm4.1}$ | $56.4_{\pm3.4}$ |
| NAGphormer | $72.5_{\pm2.3}$ | $89.9_{\pm10.4}$ | $66.5_{\pm5.6}$ | $75.1_{\pm4.3}$ | $51.7_{\pm3.5}$ |
| Specformer | $70.9_{\pm6.0}$ | $96.3_{\pm5.3}$ | $\underline{82.9}_{\pm4.9}$ | $\underline{86.6}_{\pm2.7}$ | $\underline{58.5}_{\pm3.9}$ |
| PolyFormer | $70.1_{\pm2.8}$ | $91.0_{\pm5.2}$ | $78.6_{\pm5.4}$ | $76.7_{\pm3.6}$ | $56.1_{\pm3.5}$ |
| GrokFormer | $\underline{78.2}_{\pm4.6}$ | $\mathbf{99.5}_{\pm1.6}$ | $\mathbf{94.8}_{\pm6.5}$ | $\mathbf{88.5}_{\pm5.8}$ | $\mathbf{62.2}_{\pm4.3}$ |

frequency) and largest (high-frequency) eigenvalues. More detailed settings can be found in Appendix A.2.

**Results.** The results are reported in Table 2, which shows the superiority of GrokFormer over state-of-the-art baselines in both homophilic and heterophilic datasets.

Both spatial-based and spectral-based models perform well on the homophilic networks as they can easily capture the similarity information between neighbors, and the low-pass filter is easy to fit in the homophilic networks. Although some GT-based methods, like GraphGPS, also capture the similarity information between neighbors by integrating GNNs with Transformers, resulting in certain performance improvement compared to vanilla Transformer, their heavy

Table 4: Filter fitting results in the form of SSE $\downarrow$ ($R^2$ score $\uparrow$). Lower SSE (higher $R^2$) indicates better performance.

| Models | Low-pass | High-pass | Band-pass | Band-rejection | Comb | Low-comb |
|---|---|---|---|---|---|---|
| | $\exp(-10\lambda^2)$ | $1-\exp(-10\lambda^2)$ | $\exp(-10(\lambda-1)^2)$ | $1-\exp(-10(\lambda-1)^2)$ | $\lvert sin(\pi\lambda)\rvert$ | $h_\delta(\lambda)$ |
| GCN | 3.5149(.9872) | 68.6770(.2400) | 26.2434(.1074) | 21.0127(.9440) | 49.8023(.3093) | 31.1371(.9158) |
| GAT | 2.6883(.9898) | 21.5288(.7447) | 13.8871(.4987) | 12.9724(.9643) | 22.0646(.6998) | 29.2842(.9270) |
| ChebyNet | 0.8284(.9973) | 0.7796(.9902) | 2.3071(.9100) | 2.5455(.9934) | 4.0355(.9455) | 5.0966(.9866) |
| GPRGNN | 0.4378(.9983) | 0.1046(.9985) | 2.1593(.8952) | 4.2977(.9894) | 4.9416(.9283) | 8.6554(.9768) |
| BernNet | 0.0319(.9999) | 0.0146(.9998) | 0.0388(.9984) | 0.9419(.9973) | 1.1073(.9853) | 4.5643(.9878) |
| Specformer | 0.0015(.9999) | 0.0029(.9999) | 0.0010(.9999) | 0.0027(.9999) | 0.0062(.9999) | 0.0283(.9998) |
| GrokFormer | **0.0011(.9999)** | **0.0012(.9999)** | **0.0004(.9999)** | **0.0024(.9999)** | **0.0021(.9999)** | **0.0029(.9999)** |

emphasis on global information and dependence on a freely learned attention matrix often make them susceptible to overfitting.

Heterophilic networks usually require complicated filters that have their spectrum adaptive over all eigenvalues to perform well. Thus, only Specformer and our GrokFormer can learn and fit such complex filters; the other methods fail to perform satisfactorily. Compared to Specformer, our GrokFormer performs better because our filter leverages both order- and spectral-adaptive learning power, while Specformer ignores the order-adaptive. In addition, Specformer learns node representations solely from the spectral domain, whereas our GrokFormer takes into account both spectral and spatial information simultaneously. Besides, GrokFormer can scale up in large graphs via the use of efficient self-attention and truncated decomposition.

## 5.2. Performance for Graph Classification

**Dataset.** We also conduct graph classification experiments on five TU benchmarks from diverse domains. They include three bioinformatics graph datasets, *i.e.*, PROTEINS (Borgwardt et al., 2005), PTC-MR (Toivonen et al., 2003), and MUTAG (Debnath et al., 1991) and two social network datasets, *i.e.*, IMDB-BINARY and IMDB-MULTI (Yanardag & Vishwanathan, 2015) (see Appendix A.1 for more details).

**Baselines and Settings.** We compare GrokFormer with diverse comepting models, including kernel-based methods: GK (Shervashidze et al., 2009), WL kernel (Shervashidze et al., 2011) and DGK (Yanardag & Vishwanathan, 2015), popular GNN-based models: DGCNN (Zhang et al., 2018), GCN, GIN (Xu et al., 2018), GDN (Zhao et al., 2020), and HiGCN, as well as GTs. We follow the same evaluation protocol of InfoGraph (Sun et al., 2020) to conduct a 10-fold cross-validation scheme and report the maximum average validation accuracy across folds (see Appendix A.2).

**Results.** The performance of graph classification is presented in Table 3. We find that the proposed GrokFormer outperform state-of-the-art baselines on 4 out of 5 datasets and achieve 11.9% relative improvement in PTC-MR. In

addition, compared to the kernel-based models, our approaches achieve a greater improvement, with a maximum improvement of 34.9% in PTC-MR. GNNs and GTs generally perform better than traditional kernel methods. Remarkably, due to its more expressive power, GrokFormer shows consistent superiority over the strongest baseline, Specformer, achieving an average improvement of 5.6% across all datasets.

## 5.3. Effectiveness of GrokFormer in Learning Pre-defined and Unknown Filter Patterns

### 5.3.1. Pre-defined Filters in Synthetic Datasets

Following prior work (Bo et al., 2023), we generate datasets with six filter patterns of various levels of difficulty. Specifically, images with a resolution of $100 \times 100$ from the Image Processing in Matlab library[1] are taken, with the image represented as a 2D regular grid graph with 4-neighborhood connectivity. The pixel values serve as node signals ranging from 0 to 1. These image share the same adjacency matrix $\mathbf{A} \in \mathbb{R}^{10000 \times 10000}$ and the $m$-th image has its graph signal $x_m \in \mathbb{R}^{10000}$. We apply six different predefined filters to the spectral domain of its signal, with each filter detailed in Table 4, where $h_\delta(\lambda)$ of Low-comb is defined as $I_{[0,0.5]}(\lambda) + |sin(\pi\lambda)| I_{(0.5,1)} + |sin(2\pi\lambda)| I_{[1,2]}$, with $I_\Omega = 1$ when $\lambda \in \Omega$, $I_\Omega = 0$ otherwise.

We compare the capability of our GrokFormer filter with six baselines, including GCN, GAT, ChebyNet, GPRGNN, BernNet, and Specformer, in fitting these pre-defined filter patterns through a node regression task. The hyperparameters for our model were probed in $M \in \{16, 32, 64, 128, 256\}$, $K \in \{1, 2, \cdots, 10\}$. For baselines, we set the hyper-parameters suggested by their authors and tune the hidden dimensions to maintain a consistent parameter scale. Two popular evaluation criteria – the sum of squared errors and the $R^2$ score – are used.

The fitting results are reported in Table 4. We can observe that (1) Our GrokFormer filter consistently achieve the best performance in both metrics. For complex graph filters, such

---

[1]https://ww2.mathworks.cn/products/image.html

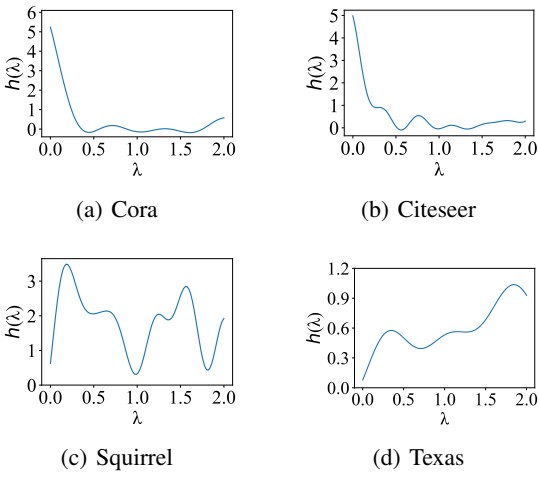

Figure 3: Filters learned by our GrokFormer on Cora and Citeseer (homophilic graphs), and Squirrel and Texas (heterophilic graphs). See Appendix B.2 for the other datasets.

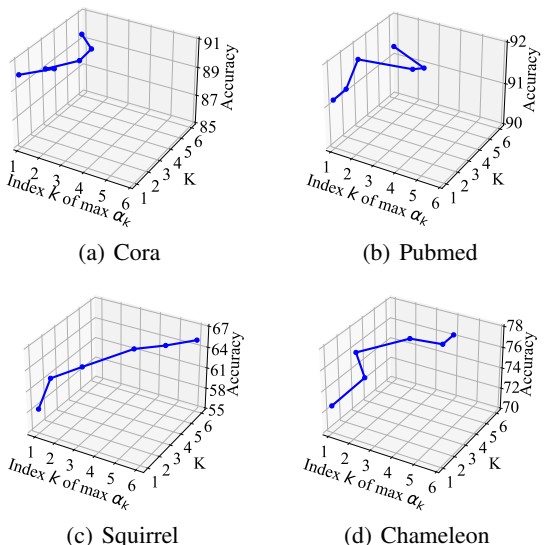

Figure 4: Order adaptivity analysis results on two homophilic graphs, including Cora and Pubmed, and two heterophilic graphs including Squirrel and Chameleon. See Appendix B.3 for the other datasets.

as Comb and Low-comb, our filter can also perform very well, demonstrating its strong expressivity in fitting the complex filters. (2) GCN and GAT can only learn low-pass filter well, which is not effective in heterophilic graph learning. (3) Polynomial-based spectral GNNs, ChebyNet, GPRGNN, and BernNet perform better than GCN by approximating graph filtering using order-adaptive polynomials. However, their expressiveness is still limited in learning complex filters due to fixed filter bases. (4) Specformer learns filters through spectrum adaptation, possessing stronger expressive ability than polynomial filters, but it is still weaker than our filter in fitting higher-order patterns as it is pre-fixed to 1st order spectrum, leading to less effective performance in fitting Comb and Low-comb. Visualization of the filter fitting results is provided in Appendix B.1.

### 5.3.2. UNKNOWN FILTERS IN REAL-WORLD HOMOPHILIC AND HETEROPHILIC DATASETS

We investigate the filters learned by GrokFormer on real-world homophilic and heterophilic datasets used in Table 2. Note that no exact ground truth filter patterns are known on these real-world datasets, but the visualization of the learned spectrum offers important insights into how the spectrum and order adaptability in GrokFormer enables the learning of complex homophily/heterophily relations.

**Adaptability in Graph Spectrum.** To analyze the importance of spectrum adaptability, we plot the filters learned by GrokFormer on two typical homophilic datasets, Cora and Citeseer, and two heterophilic datasets, Squirrel and Texas, in Figure 3. We observe that GrokFormer learns a low-pass filter on Cora and Citeseer, aligning with their strong homophily. In contrast, on Texas, which exhibits strong

heterophily, GrokFormer adaptively learns a high-pass filter to capture the differential information between nodes. On Squirrel, which has dense and mostly heterophilic edges, GrokFormer also effectively learns the heterophily, resulting in a comb-alike filter. Importantly, GrokFormer does not require prior knowledge to manually tune the spectrum hyperparameters to achieve this; it learns to adaptively fit different filters hidden in the graphs with different homophilic and heterophilic properties. Similar results can be found for the other datasets in Appendix B.2.

**Adaptability in Graph Spectral Order.** Similarly, we also analyze the order adaptability of GrokFormer on Cora, Pubmed, Squirrel and Chameleon. To this end, we evaluate the accuracy performance of GrokFormer with all other settings fixed except that we increase the order $K$ from one to six, from which we observe the relation of $k$ and $K$ w.r.t. the best accuracy. The results are shown in Figure 4. Cora and Pubmed are strong homophilic network that expect the low-pass filter, and such filter is easy to learn, on which GrokFormer fits the graph adaptively with a small $K$ (*i.e.*, $K \leq 3$) rather than overfitting it with a large $K$. Squirrel and Chameleon have a large number of heterophilic edges, requiring a more complex frequency response (see Figure 3), on which GrokFormer learns to adaptively use a large $K$ (*i.e.*, $K > 3$) for capturing rich frequency components instead of restricting in using small $K$ values. Additionally, we can observe that the maximum order coefficient $\alpha_k$ is distributed within the $K \leq 3$ order filter basis on Cora and Pubmed, while on Squirrel and Chameleon, as the order $K$ increases, the largest order coefficient $\alpha_k$ extends to

Table 5: Ablation studies on node- and graph-level tasks

| Method | Node-level | | | Graph-level | |
|---|---|---|---|---|---|
| | Cora | Penn94 | Texas | PROTEINS | IMDB-B |
| SE | $77.42_{\pm1.77}$ | $76.29_{\pm0.35}$ | $90.33_{\pm2.36}$ | $71.4_{\pm3.7}$ | $74.9_{\pm3.8}$ |
| GFKAN | $88.98_{\pm1.25}$ | $81.36_{\pm0.29}$ | $93.62_{\pm3.04}$ | $76.1_{\pm3.5}$ | $87.3_{\pm2.5}$ |
| Full Model | $89.57_{\pm1.43}$ | $83.59_{\pm0.26}$ | $94.59_{\pm2.08}$ | $78.2_{\pm4.6}$ | $88.5_{\pm5.8}$ |

the higher-order filter basis. This shows that our method can learn to adaptively assign a large weight to the most suitable filter bases, thereby achieving a synthesized filter specifically for the training graph. More results can be found for the other datasets in Appendix B.3.

### 5.4. Ablation Study

The ablation study is performed to analyze the performance of GrokFormer (Full Model) compared to its two variants: i) **Self-attention-E** (SE), which contains only efficient self-attention mechanism in GrokFormer, with the proposed spectral graph convolution module removed; ii) **Graph Fourier KAN** (GFKAN) that keeps the spectral graph convolution module only. The results on three node classification datasets and two graph classification datasets are reported in Table 5. It can be observed that Self-attention-E shows good performance in some datasets such as Texas, outperforming most spatial methods in Table 2, due to its ability to capture the feature similarity of the global node. However, it struggles to perform well on the other graph datasets due to its limitation in capturing non-low frequency graph information. The proposed Graph Fourier KAN significantly enhances the performance, showing competitive performance against the state-of-the-art competing methods in Table 2. This superiority benefits from its order and spectrum adaptability that enables expressive graph representation learning without using self-attention. However, on the large-scale dataset penn94, which exhibits a low level of homophily, relying solely on spectral information from Graph Fourier KAN proves insufficient. GrokFormer achieves consistently improved performance only when both Graph Fourier KAN and self-attention are integrated, demonstrating its ability to effectively synthesize the strengths of both modules and outperform its individual variants.

### 5.5. Empirical Time and Space Complexities

In this section, we apply Cora and Penn94 to verify the efficiency of GrokFormer. We test the time and space overheads of GrokFormer and two spectral GTs, *i.e.*, PolyFormer and Specformer.

**Setup.** To perform a fair comparison, we run each model for 1,000 epochs and report the total time and space costs. We set the hidden dimension $d = 64$ for all methods. For Specformer and our GrokFormer, we use use full eigenvectors for Cora and 6,000 eigenvectors for Penn94. We set $K = 10$

Table 6: The training cost in terms of GPU memory (MB) and running time (s).

| Dataset | Method | Memory (MB) | Time (s) |
|---|---|---|---|
| | PolyFormer | 1836 | 13.58 |
| Cora | Specformer | 1509 | 4.35 |
| | GrokFormer | 1267 | 3.23 |
| | PolyFormer | 14113 | 121.78 |
| Penn94 | Specformer | 5053 | 9.39 |
| | GrokFormer | 4647 | 8.13 |

for polynomial bases of PolyFormer. The pre-processing of all models is not included in the training time. The results are shown in Table 6.

**Results.** From Table 6, we can find that our GrokFormer shows high efficiency. Compared with Specformer that performs self-attention on $N$ eigenvalues with $\mathcal{O}(N^2)$ complexity, Fourier series representation in our GrokFormer offers lower training complexity, scaling linearly with $\mathcal{O}(N)$. PolyFormer needs to calculate the self-attention weights of $K$ polynomial bases for $N$ times, which requires a lot of computations, but Specformer and our GrokFormer only need to calculate $\mathbf{U}diag(\lambda)\mathbf{U}^\top\mathbf{X}$ once for non-local information passing. Besides, Specformer and our GrokFormer utilize a 2-layer MLP (embedding layer) on the feature matrix $\mathbf{X}$ to reduce the feature dimension $F$ to number of classes $C$ ($C \ll F$) before information passing, but PolyFormer does not have this embedding layer, so it consumes more computation. In the large graph Penn94, Specformer and our GrokFormer use truncated spectral decomposition to reduce the forward complexity, so it is more efficient than PolyFormer.

## 6. Conclusion and Future Work

This paper presents GrokFormer, a novel GT model that learns expressive, adaptive spectral filters through order-$K$ Fourier series modeling, overcoming the limitations of self-attention to effectively capture rich frequency signals in a broad range of graph spectrum and order. Experiments on the synthetic dataset show that the proposed GrokFormer filter is more expressive than SOTA graph filters used in GNNs and GTs. Comprehensive experiments on real-world datasets also demonstrate that the superiority of GrokFormer over SOTA GNNs an GTs. A promising future direction is to learn graph filters with strong expressiveness more efficiently, without the need for spectral decomposition.

## Acknowledgments

This research is supported in part by A*STAR under its MTC YIRG Grant (No. M24N8c0103), the Ministry of Education of Singapore under its Tier-1 Academic Research Fund (No. 24-SIS-SMU-008), and the Lee Kong Chian Fellowship.

## Impact Statement

This paper presents work whose goal is to advance the field of graph machine learning. There are many potential societal consequences of our work, none of which we feel must be specifically highlighted here.

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

# A. EXPERIMENTAL DETAILS

## A.1. DATASETS

Table 7: Statistics of node classification datasets.

| Datasets | Cora | Citeseer | Pubmed | Photo | WikiCS | Physics | Penn94 | Chameleon | Squirrel | Actor | Texas |
|---|---|---|---|---|---|---|---|---|---|---|---|
| #Nodes | 2,708 | 3,327 | 19,717 | 7,650 | 11701 | 34,493 | 41,554 | 2,277 | 5,201 | 7,600 | 183 |
| #Edges | 5,429 | 4,732 | 44,338 | 238,163 | 216,123 | 247,962 | 1,362,229 | 36,101 | 217,073 | 33,544 | 295 |
| #Features | 1,433 | 3,703 | 500 | 745 | 300 | 500 | 4,814 | 2,325 | 2,089 | 931 | 1,703 |
| #Classes | 7 | 6 | 3 | 8 | 10 | 5 | 2 | 5 | 5 | 5 | 5 |
| $\mathcal{H}$ | 0.81 | 0.74 | 0.80 | 0.83 | 0.57 | 0.91 | 0.47 | 0.23 | 0.22 | 0.22 | 0.06 |

The homophily ratio $\mathcal{H}$ in Table 7 as a measure of the graph homophily level is used to define graphs with strong homophily/heterophily. The homophily ratio is defined as $\mathcal{H} = \frac{|\{(v_i,v_j):(v_j,v_i)\in\mathcal{E}\wedge y_i=y_j\}|}{|\mathcal{E}|}$ (Zhu et al., 2020), which is the fraction of edges in a graph which connect nodes that have the same class label. Homophily ratio $\mathcal{H} \to 1$ represents the graph exhibit strong homophily, while the graph with strong heterophily (or low/weak homophily) have small homophily ratio $\mathcal{H} \to 0$.

Table 8: Statistics of graph classification datasets

| Datasets | PROTEINS | MUTAG | PTC-MR | IMDB-B | IMDB-M |
|---|---|---|---|---|---|
| #Graphs | 1113 | 188 | 344 | 1000 | 1500 |
| #Classes | 2 | 2 | 2 | 2 | 3 |
| #Nodes (Max) | 620 | 28 | 109 | 136 | 89 |
| #Nodes (Avg.) | 39.06 | 17.93 | 14.29 | 19.77 | 13.00 |
| #Edges (Avg.) | 72.82 | 19.79 | 14.69 | 13.06 | 65.93 |

## A.2. DETAILED EXPERIMENTAL SETUP

### A.2.1. OPERATING ENVIRONMENT

For the implementation, we utilize NetworkX, Pytorch, and Pytorch Geometric for model construction. All experiments are conducted on NVIDIA GeForce RTX 3090 GPUs with 24 GB memory, TITAN Xp GPU machines equipped with 12 GB memory.

### A.2.2. NODE CLASSIFICATION

We train all models with the Adam optimizer (Diederik & Ba, 2015) following previous works (Bo et al., 2021; 2023). We run the experiments with 2,000 epochs and stop the training in advance if the validation loss does not continuously decrease for 200 epochs. Classification accuracy is used as a metric to evaluate the performance of all models (Kipf & Welling, 2017; Veličković et al., 2018). For the large-scale dataset Penn94, (Lim et al., 2021) provides five official splits, so we run it five times to report the mean accuracy. For other datasets, we run the experiments ten times, each with a different random split. Moreover, due to the increased number of nodes and edges, we set $K = 10$ for Penn94.

The hyper-parameter ranges we used for tuning on each dataset are as follows:

- Number of layers: $\{1, 2, 3\}$;

- Number of Fourier series expansion terms: $\{16, 32, 64\}$;

- Number of heads: $\{1, 2, 3, 4, 5\}$;

- Hidden dimension: $\{64, 128\}$;

- Learning rate: $\{0.01, 0.005\}$;

- Number of K: $\{1, 2, 3, 4, 5, 6, 10\}$;

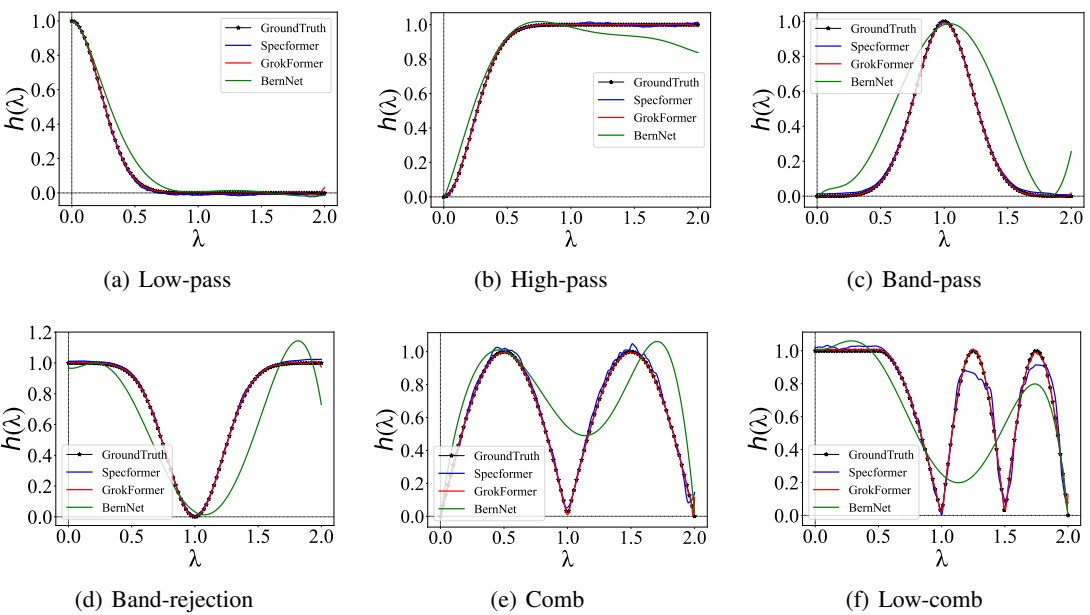

Figure 5: Illustrations of six filters and their approximations learned by our GrokFormer filter, BernNet, and Specformer.

- Weight decays: {5e-3, 5e-4, 5e-5};

- Dropout rates: {0.0, 0.1, 0.2, 0.3, 0.4, 0.5, 0.6, 0.7, 0.8}.

### A.2.3. GRAPH CLASSIFICATION

We use the Adam optimizer to train all models. Following (Sun et al., 2020), we perform 10-fold cross validation. We report the average and standard deviation of validation accuracies across the 10 folds within the cross-validation. We implement readout operations by conducting max pooling to obtain a global embedding for each graph. Since the computational complexity of vanilla self-attention in graph classification task can be alleviated by tuning batch size, we employ the vanilla self-attention mechanism in the graph-level representation learning. Hyperparameter selection range is as follows:

- Number of layers: {1, 2};

- Epoch: {100, 200, 300};

- Learning rates: {0.01, 0.005, 0.001};

- Weight decay: {0.0, 0.0005, 0.00005};

- Dropout rate: {0.0, 0.05, 0.1};

- Number of Fourier series expansion terms: {16, 32, 64};

- Hidden dim: {32, 64, 128};

- Number of K: {1, 2, 3, 4, 5, 6};

- Batch size: {128};

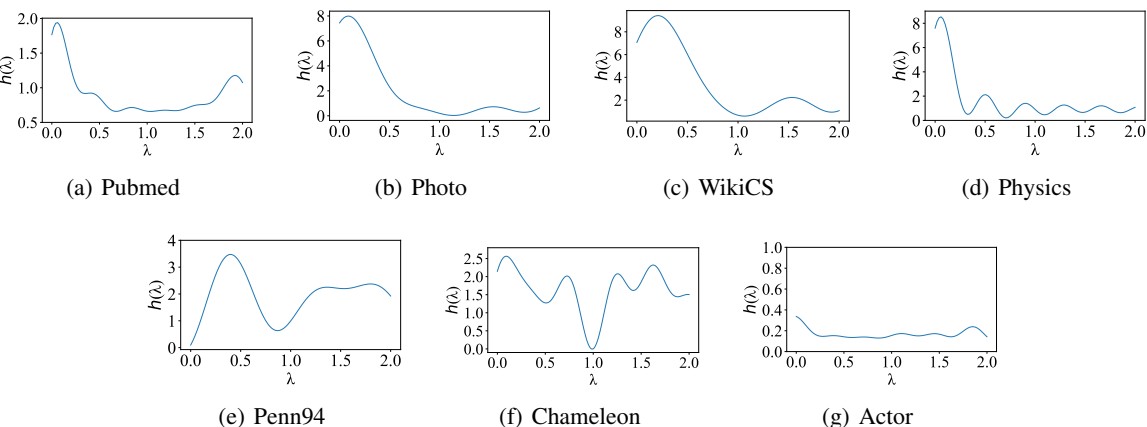

Figure 6: Filters learned from real-world datasets with varying graph properties by our GrokFormer.

## B. MORE EXPERIMENTAL RESULTS

### B.1. VISUALIZATION OF LEARNED FILTERS ON SYNTHETIC DATASETS

Here we illustrate six filters and their approximations learned by our GrokFormer filter, BernNet, and Specformer in Figure 5. In general, GrokFormer filter can learn a precise approximation of these filters. However, polynomial filter of BernNet is difficult to fit these filters, especially other complex filters beyond low-pass and high-pass filters. Although Specformer has been well fitted, our proposed filter can perform much better, especially on filters with more complex patterns, such as Comb and Low-comb.

### B.2. VISUALIZATION OF LEARNED FILTERS ON REAL-WORLD DATASETS

In this section, we visualize more filter results learned by our GrokFormer on real-world datasets. From Figure 6, we find that (1) on homophilic graphs, our proposed filter learns low-pass filters with different amplitude and frequency responses for them, which is consistent with the homophily property, *i.e.*, the low-frequency information is important in the homophilic scenario. (2) Edges in Chameleon and Penn94 datasets are dense and mostly heterophilic, so our proposed filter learns comb-alike filters with complex frequency components for them. (3) our GrokFormer filter learns an all-pass filter on the Actor dataset protecting its raw features, which is consistent with the fact that its raw features are associated with labels (He et al., 2021).

### B.3. ADAPTIVITY IN GRAPH SPECTRAL ORDER

Figure 7 shows additional results on order adaptivity. We observe that GrokFormer achieves the best performance when $K$ is small on all homophilic datasets. This is because the low-pass filters desired by homophilic networks (see Figure 6) are easy to learn. GrokFormer fits the graph adaptively with a small $K$ rather than overfitting it with a large $K$. In addition, as shown in Figure 3 and Figure 6, a high-pass filter required by the strong heterophilic network Texas, and an all-pass filter desired by the Actor are also easy to learn, so GrokFormer fits them adaptively with a small $K$. However, for Penn94, which have dense edges and predominantly heterophilic, complex comb-like filters (see Figure 6) are required. As a result, GrokFormer learns to adaptively use a larger $K$ to capture a broader range of frequency components, rather than restricting itself to a small $K$. Moreover, we can find that GrokFormer filter assign the largest order coefficient $a_k$ to $K \leq 3$ order filter basis on these datasets that expect simple filters (low-pass, all-pass, high-pass), while on Penn94, the largest order coefficient $a_k$ extends to the higher-order filter basis. This demonstrates that our method can effectively learn order-adaptive filters for datasets with varying properties.

### B.4. GRAPH CLASSIFICATION AND REGRESSION

In this section, we conducts experiments on additional graph-level datasets, including a subset (12K) of ZINC molecular graphs (250K) dataset (Irwin et al., 2012), the super-pixels dataset CIFAR10 (Dwivedi et al., 2023), and a long-range graph

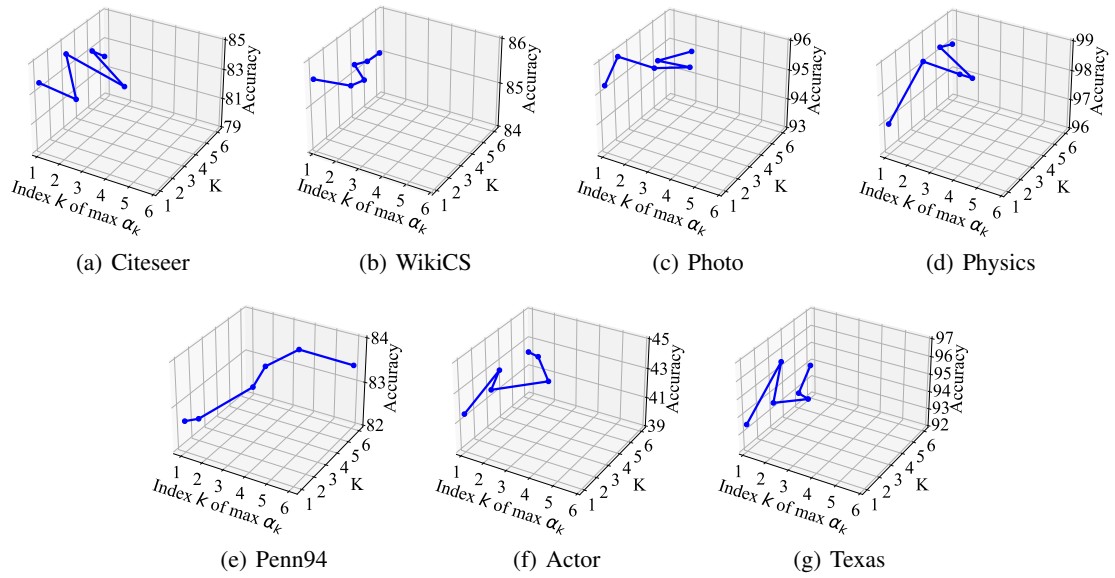

Figure 7: Order adaptivity analysis results.

Table 9: Detailed information of additional graph-level datasets.

|  | #Graphs | Avg. #nodes | Avg. #edges | Node feat. (dim) | Edge feat. (dim) | Tasks | Metric |
|---|---|---|---|---|---|---|---|
| ZINC | 12,000 | 23.2 | 24.9 | Atom Type (28) | Bond Type (4) | Regression | MAE |
| CIFAR10 | 60,000 | 117.6 | 941.1 | Pixel[RGB]+Coord (5) | Node Dist (1) | Classification | ACC |
| Peptides-func | 15,535 | 150.9 | 307.3 | Atom Encoder (9) | Bond Encoder (3) | Classification | AP |

benchmark Peptides-func (Dwivedi et al., 2022). We choose graph Transformers with positional or structural embedding (Graphormer, GraphGPS and GRIT) as the baselines. All experiments are conducted on the standard train/validation/test splits of the evaluated benchmarks. We use the hyperparameter for the baselines as suggested in their respective papers. Our hyper-parameters and ranges were as follows:

- Dropout: {0.0, 0.05, 0.1};

- Number of layers: {4, 6, 8};

- Number of Fourier series expansion terms: {16, 32, 64};

- Number of heads: {1, 2, 3, 4, 5};

- Learning rate: {0.001, 0.0001, 0.0005};

- Number of K: {1, 2, 3, 4, 5, 6};

- Weight decays: {5e-4, 5e-5};

Table 10: Results on additional graph-level datasets. '∗' means edge feature is not encoded.

|  | ZINC (MAE↓) | CIFAR10 (ACC↑) | Peptides-func (AP↑) | Peptides-func* |
|---|---|---|---|---|
| Graphormer | 0.122 | - | - | - |
| GraphGPS | 0.070 | 72.31 | 0.6535 | 0.6257 |
| GRIT | 0.060 | 75.67 | 0.6988 | 0.6458 |
| GrokFormer | 0.076 | 74.26 | 0.6415 | 0.5987 |

- Internal MPGNN: {GCN, GatedGCN(Bresson & Laurent, 2017)};

From the results, we observe that GrokFormer achieves better performance than Graphormer and performs comparably well to GraphGPS. GrokFormer slightly underperforms GRIT on ZINC and CIFAR10, while the performance margin on Peptides-func is relatively large. GRIT is designed to improve GT's expressiveness in large datasets by incorporating graph inductive biases, so it shows an advantage on long-range graph dataset. Although GraphGPS achieves better performance on ZINC and Peptides-func, it exhibits suboptimal results on CIFAR10 and the node-level datasets due to overfitting. In GrokFomer, we aim to enhance the GT's capability to capture various frequency information on graphs by designing an expressive filter. It achieves a relatively better balance between the generalization ability and the expressiveness, leading to a robust performance on both node-level and graph-level datasets.

## C. THEORETICAL PROOFS

In the following, we present the proof for **Proposition 4.1**.

*Proof.* For the spectrum of graph Laplacian $\lambda$, the corresponding arbitrary order is given by $[\lambda^0; \lambda^1; \lambda^2; \cdots; \lambda^K]$. When processed by the order-wise MLP with the trainable weight $\mathbf{w} = [w_0, w_1, \cdots, w_K] \in \mathbb{R}^{1 \times K}$, the new spectrum $\lambda_{new}$ is updated as $\lambda_{new} = w_0 \lambda^0 + w_1 \lambda^1 + w_2 \lambda^2 + \cdots + w_K \lambda^K$.

In Eq. (9), we eliminate the learnable nonlinear function over the spectrum and define our GrokFormer filter function as follows:

$$h(\lambda) = \sum_{k=0}^{K} \alpha_k \lambda^k, \tag{13}$$

where $\alpha_k$ is learned based on the $k$-order spectrum $\lambda^k$. This value serves as order-adaptive weight for the polynomial filter similar to $w_k$ of MLP. Therefore, the designed GrokFormer filter function is learnable in terms of the spectral order.

Secondly, GrokFormer filter function can be reduced into a simpler form as follows, when removing the order adaptivity term and the higher-order term:

$$h(\lambda) = \sum_{m=0}^{M} \left( \cos(m\lambda) \cdot a_m + \sin(m\lambda) \cdot b_m \right). \tag{14}$$

Here, $sin(m\lambda)$ and $cos(m\lambda)$ with $m \in [0, M]$ scale the spectrum with different frequency components. Therefore, different scales of the spectrum are adjustable due to the presence of learnable coefficient $a_m$ and $b_m$. They serves as the spectrum-adapted weights for the filter. Therefore, GrokFormer filter function is also learnable in terms of the graph spectrum. □

In the following, we present the proof for **Proposition 4.2**.

Table 11: The filter form of polynomial GNNs

| Model | Filter |
|---|---|
| APPNP | $h(\lambda) = \sum_{k=0}^{K} \frac{\gamma^k}{1-\gamma}(1-\lambda)^k$ |
| GPR-GNN | $h(\lambda) = \sum_{k=0}^{K} \gamma_k (1-\lambda)^k$ |
| BernNet | $h(\lambda) = \sum_{k=0}^{K} \alpha_k \binom{K}{k}(1-\frac{\lambda}{2})^{K-k}(\frac{\lambda}{2})^k$ |
| JacobiConv | $h(\lambda) = \sum_{k=0}^{K} \alpha_k \sum_{s=0}^{k} \frac{(k+a)!(k+b)!(-\lambda)^{k-s}(2-\lambda)^s}{2^k s!(k+a-s)!(b+s)!(k-s)!}$ |

*Proof.* Polynomial filters are popular in graph representation learning. We show below that the state-of-the-art (SOTA) polynomial filters listed in Table 11 is a simplified form of our proposed filter, some of which are utilized by SOTA methods such as FeTA (Bastos et al., 2022) and PolyFormer (Ma et al., 2024).

First of all, the polynomial filter functions in Table 11 can be uniformly written as follows:

$$h(\lambda) = \alpha_0 + \alpha_1\lambda + \alpha_2\lambda^2 + \cdots \alpha_K\lambda^K = \sum_{k=0}^{K} \alpha_k\lambda^k, \tag{15}$$

where $\alpha$ is a learnable parameter. These polynomial functions have fixed filter bases (*i.e.*, $\lambda, \lambda^2, \cdots, \lambda^K$), which approximate arbitrary spectral filters in an order-adaptive manner. According to the above Proof for Proposition 4.1, our proposed filter can be simplified to:

$$h(\lambda) = \sum_{k=0}^{K} \alpha_k\lambda^k. \tag{16}$$

Therefore, these polynomial filters are the case of a simplified variant of our GrokFormer filter. $\qquad\square$

Below, we present the proof for **Proposition 4.3**.

*Proof.* The very recent model Specformer (Bo et al., 2023) learns graph filters $h_s(\lambda)$ via eigenvalue encoding, which can be treated as a linear combination of position encoding in graph Transformer when the self-attention matrix is set to the identity matrix:

$$h_s(\lambda) = a_0\lambda + \sum_{i=1}^{d/2} a_i sin(\frac{\epsilon\lambda}{10000^{2i/d}}) + \sum_{i=1}^{d/2} b_i cos(\frac{\epsilon\lambda}{10000^{2i/d}}), \tag{17}$$

where $\epsilon$ is a hyperparameter and $d$ is the dimension. $a_i$ and $b_i$ are learnable parameters. If $M = d/2$ and $m = \frac{\epsilon}{10000^{i/M}}$ are used, Eq. (17) can be rewritten as follows:

$$h_s(\lambda) = a_0\lambda + \sum_{i=1}^{M} (\sin(m\lambda) \cdot a_i + \cos(m\lambda) \cdot b_i). \tag{18}$$

Therefore, the Specformer filter learns over the specific first-order spectrum of graph Laplacian.

In Eq. (18), the term $a_0\lambda$ can be combined into sine and cosine terms in an approximate manner. Suppose that constants R and $\phi$ can be found such that:

$$a_0\lambda = R\sin(\lambda + \phi). \tag{19}$$

We can approximate $a_0\lambda$ as a linear combination of $R\sin(\lambda + \phi)$. Given that the sine function has the linear combination form $\sin(x + \phi) = \sin(x)\cos(\phi) + \cos(x)\sin(\phi)$, we can get the following:

$$a_0\lambda = R(\sin(\lambda)\cos(\phi) + \cos(\lambda)\sin(\phi)), \tag{20}$$

where $R, \cos(\phi), \sin(\phi)$ are constants. Therefore, Eq. (18) can be rewrite as follows,

$$h_s(\lambda) = \theta_1\sin(\lambda) + \theta_2\cos(\lambda) + \sum_{i=2}^{M} (\sin(m\lambda) \cdot a_i + \cos(m\lambda) \cdot b_i), \tag{21}$$

where $\theta_1 = R\cos(\phi)a_1$ and $\theta_2 = R\sin(\phi)b_1$.

According to the above Proof for Proposition 4.1, our GrokFormer filter can be written as follows:

$$h(\lambda) = \sum_{m=0}^{M} (\sin(m\lambda) \cdot a_m + \cos(m\lambda) \cdot b_m). \tag{22}$$

We can further write the Eq. (22) in the following form:

$$h(\lambda) = a_0 + a_1\sin(\lambda) + b_1\cos(\lambda) + \sum_{i=2}^{M} (\sin(i\lambda) \cdot a_i + \cos(i\lambda) \cdot b_i). \tag{23}$$

Comparing Eq. (21) and Eq. (23), it is clear that the Specformer filter is a simplified variant of our GrokFormer filter.

$\qquad\square$

Next, we provide the proof for **Proposition 4.4**.

*Proof.* According to the uniform convergence of Fourier series (Stein & Shakarchi, 2011), for any continuous real-valued function $f(x)$ on [a,b] and $f'(x)$ is piece-wise continuous on [a,b] and any $\epsilon > 0$, there exists a Fourier series $P(x)$ converges to $f(x)$ uniformly such that

$$\max_{a \leq x \leq b} |P(x) - f(x)| < \epsilon. \tag{24}$$

Since the eigenvalues fall in the range [0, 2] and our GrokFormer filter is constructed from Fourier series representation, our filter can approximate any continuous function in the interval [0, 2] based on the uniform convergence of Fourier series above.

Secondly, the spectral graph convolution in Eq. (8) based on the learnable filter is permutation equivariant because $(\mathbf{PUP}^\top)(\mathbf{P\Lambda P}^\top)(\mathbf{PUP}^\top)^\top = \mathbf{P}(\mathbf{U\Lambda U}^\top)\mathbf{P}^\top$, given an arbitrary permutation matrix $\mathbf{P}$.

$\square$

