# OpenReview forum: "GrokFormer: Graph Fourier Kolmogorov-Arnold Transformers"
_ICML.cc/2025/Conference — ICML 2025 poster_

### Official Review · Reviewer_tPj2 · 2025-03-03

**Overall Recommendation:** 3

**Summary:**

This paper propose to introduce a novel Kolmogorov-Arnold networks (KAN) based spectral filters to graph Transformer framework to enhance the flexibility to perform low/high/band-pass filtering.
Compared to the previous polynomial spectral graph neural networks (GNNs) as well as the proposed graph KAN spectral filter has orthogonal polynomial bases, better convergence as well as global graph modeling.
Compared to the non-polynomial spectral GNNs, Specformer, the proposed spectral filter can consider the higher-order of the eigenvalues in an explicit manner, leading to more flexible construction.
The authors also show empirically that adding an linear-attention module to the spectral filters can lead to better empirical performance.


## update after rebuttal

I have updated the scores according to the rebuttal of the authors. Details please see the rebuttal comments

**Claims And Evidence:**

1. The authors claim that Specformer cannot consider spectrum at different orders while the proposed method can.
- This is not well justified since in Specformer, there are MLP applied to each eigenvalue $\lambda_i$ which can learn to output $\lambda_i^k$ for $k>0$ theoretically, given by the universality of MLP. Therefore, this claim is problematic.


2. The authors claim that self-attention preserves only low-frequency signals in graph features in abstract.
- This is not well supported. It is true that each attention map is equivalent to a low-pass filter. However, in multi-head attentions, a linear combinations of different low-pass filters and all-pass filters (i.e., residual connection) might generate band/high-pass filters.


3. The authors claim to propose a novel graph Transformers, which is more flexible and expressive.
- First, the proposed networks is more like a spectral GNNs plus a linear-attention. The "transformer" component is not well justified. (The differences with/without attention in the ablation studies are not even staistically significant from T-test).
- There is no theoretical and empirical support that the proposed model is more expressive than the recent graph Transformers such as GraphGPS (Rampasek et al., 2022), GD-Graphormer (Zhang et al., 2023), GRIT (Ma et al., 2023).
- In the baselines, there are some graph Transformers known for better scalability but not on top of the expressivity. The comparison to them cannot support the claim on stronger expressivity.
- Stronger ability to generate filters of different frequencies does not necessary lead to stronger expressivity on distinguish different graph-structure. Wang et al., (2022) shows that the expressivity of polynomial spectral GNNs are not more expressive than 1-WL. CKGConv (Ma et al., 2024), shows that applying non-linearity transformation on the polynomial bases can lead to stronger expressivity beyond 1-WL. There are no similar discussion in the paper to support the statement of "more expressive"



--------------
> only the references not included in the paper are listed


- Zhang, Bohang, et al. "Rethinking the Expressive Power of GNNs via Graph Biconnectivity." The Eleventh International Conference on Learning Representations.
- Ma, Liheng, et al. "Graph inductive biases in transformers without message passing." International Conference on Machine Learning. PMLR, 2023.
- Wang, Xiyuan, and Muhan Zhang. "How powerful are spectral graph neural networks." International conference on machine learning. PMLR, 2022.
- Ma, Liheng, et al. "CKGConv: General Graph Convolution with Continuous Kernels." International Conference on Machine Learning. PMLR, 2024.

**Essential References Not Discussed:**

No

**Experimental Designs Or Analyses:**

1. The ablation study cannot well justify the necessary of the efficient self-attention.
- the performance difference between the graph Fourier KAN and Full model is not statisically significant, indiciating that the efficient-self-attention is not necessary.
- consider using other datasets with stronger distinguishability on the model capacity.

**Methods And Evaluation Criteria:**

1. The datasets utilized are known for the lack of training examples, which are less suitable to evaluate the capacity of complicated models such as graph Transformers. The overfitting issues might dominate the performance improvements. These datasets can justify that the proposed method introduce better inductive bias. However, using these datasets only cannot well support the capacity statement of the proposed GrokFormer.

2. In the literature of graph Transformers, the datasets from BenchmarkingGNNs (Dwivedi et al., 2023) and Long Range Graph Benchmarks (Dwivedi et al., 2022) are more widely used to evaulate the capacity and expressivity of the graph models. (They have been shown more effeictive to evaluate the ability of modeling graph-structure compared to TU datasets). However, the paper does not utilize those datasets and weaken the experimental conclusions.

3. The proposed method claims that the proposed spectral filter can introduce the stronger capacity to graph Transformers.
However, the compared graph Transformers are more known for better scalability instead of stronger expressivity.
 Most existing expressive graph Transformers are ignored in the comparison, including but limited to GraphGPS (Rampasek et al., 2022), GD-Graphormer (Zhang et al., 2023), GRIT (Ma et al., 2023).  (listed in previous section) Epesicallly for GraphGPS and GRIT, the former one is also hybrid Transformer architecture (MPNN+self-attention) and the latter one reach SOTA performance on various datasets with pure Transformer architecture.

4. The filter fitting experiments does not include graph Transformers in comparisons.

**Other Comments Or Suggestions:**

1. In graph theory, the term ``graph order'' is widely used, which stands for the number of nodes in a graph. This work uses this term mistakenly.

2. Arguably speaking, Specformer (Bo et al., 2023) shall be considered as a spectral GNNs with Transformer/self-attention encoders rather than graph Transformers, which are supposed to learn graph-structure data directly with Transformer architecture (self-attention + FFN).

**Other Strengths And Weaknesses:**

**Weaknesses**

1. The proposal of the graph Fourier KAN spectral filters are novel and interesting. However, the injection of efficient self-attention is not well supported, which in fact weaken the paper. A simple A+B technique combination without good motivation and theoretical/empirical support  lead to a negative viewpoint.

2. No expressivity discussion on the proposed method.

**Questions For Authors:**

1. The method seems to focus on the capacity and expressivity of the model. Why not use full self-attention? What is the main goal of using the linear-attention which is theoretically weaker than the regular self-attention.
2. How necessary for the attention modules in the work?

**Relation To Broader Scientific Literature:**

1. The proposed method does not provide key contribution to the literature of graph Transformers as well as Transformers. This unmatch the main statement of the paper.
2. The key contribution of the proposed method is proposing a novel parameterization of the graph spectral filters and spectral GNNs, which endows with several desired properties.

**Theoretical Claims:**

1. Listed in the (1) of **Claims And Evidence**
2. The Complexity claim has some issues.
- When consider the full spectrum ($N$ eigenvalues/eigenvectors), the forward pass of the proposed method is actually $O(N^3)$ ($O(N^2 M)$ with $M=N$) on eq. (8), which is missed in the calculation of the overall forward complexity.
3. In table 1, there are two properties 'order-adaptive' and 'spectrum-adaptive'. These properties are actually vague. They are viewed as positive feature of the work without clear definition and support.
- In Polynomial Spectral GNNs, the *order-adaptive* is referred to the polynomial approximation to the full spectrum, which is an approximation from the spatial domain. This is a trade-off between the flexibility and the computational cost.
- In Specformer, the *spectrum-adaptive* is referred to the usage of a subset of spectrums (eigenvalues), which is an approximation from the spectral domain.
- In Graph KAN spectral filter, *order-adaptive* is referred to the usage of the $k$-power of eigenvalues. However, this is different from the *order-adaptive* in Polynomial Spectral GNNs and can neither lead to better computational cost nor leading to a tighter approximation.
- In fact, it is arguably that the proposed KAN-based spectral filter can be well approximated by the transformer-based spectral filter in the specformer due to the universal approximation. (However, the performance might be different in practical training).

---

> ### Author Rebuttal · Authors · 2025-04-01
>
> We sincerely appreciate your constructive and positive comments on the filter design. Please see our response to your comments one by one below.
>
> >**Question #1** In Specformer, MLP applied to each eigenvalue $\lambda$, which can learn to output for $\lambda^k$ theoretically.
>
> We appreciate your insightful comment. While applying MLP to $\lambda$ in Specformer can help theoretically capture frequency information about $\lambda^k$, it does not necessarily enable the MLP in Specformer to effectively model spectral information across arbitrary filter orders $k$ due to its limited parameters and the lack of explicit learning for unknown $\lambda^k$. In contrast, our GrokFormer explicitly captures a diverse range of frequencies across different filter orders, ensuring a more expressive representation in the spectral domain.
>
> >**Question #2** In multi-head attention, the residual connection might generate band/high-pass filters.
>
> We really appreciate your comments. We'd like to clarify that [Ref1] has shown that *although multi-head, FFN, and skip connection all help preserve the high-frequency signals, none would change the fact that MSA block as a whole only possesses the representational power of low-pass filters* (see page 5 of [Ref1]).  The theory is further supported by the experiments on the ZINC dataset in [Ref2]. These findings confirm that Transformer is essentially a low-pass filter, motivating our development of expressive spectral filters for graph Transformers.
>
> - [Ref1] Anti-Oversmoothing in Deep Vision Transformers via the Fourier Domain Analysis: From Theory to Practice. ICLR, 2022.
>
> - [Ref2] How Expressive are Transformers in Spectral Domain for Graphs?  TMLR, 2022
>
> >**Question #3** How necessary for the attention modules in the work? (Lack of statistical analysis from t-test).
>
> In Table A1, we include a new dataset, Penn94 in ablation study and report the $p$-Value using t-tests on all datasets.
>
> ```
> Table A1. Additional ablation study on Penn94.
> ```
> | |Penn94|$p$-Value|
> | --- | --- | --- |
> |Self-attention-E|76.29|0.001|
> |Graph Fourier KAN|81.36|0.002|
> |GrokFormer|83.59|-|
>
> Self-attention-E can generally capture global spatial relationships, preserving the similarity of node features.  However, on the large-scale dataset penn94 with a low homophily level, relying solely on spectral information from Graph Fourier KAN may not be sufficient.  Only when both Graph Fourier KAN and self-attention are integrated, GrokFormer can achieve consistently improved performance, with statistically significant differences ($p$ < 0.05) on the ablated GrokFormer models. This demonstrates that our GrokFormer is more than just a simple combination of self-attention and Graph Fourier KAN.
>
> >**Question #4** The theoretical discussion of strong expressivity on WL test.
>
> Please see our detailed response to Reviewer axD1's Weakness #3.
>
> >**Question #5** and **Question #6** Some expressive graph Transformers like GraphGPS and GRIT are overlooked in comparisons.
>
> Please see our detailed response to Reviewer axD1's Weakness #1 and #2.
>
> >**Question #7** The filter fitting experiments do not include graph Transformers in comparisons.
>
> Both Polyformer and Specformer design filter functions h($\lambda$) in the spectral domain, which can be regarded as spectral graph Transformers based on a Transformer architecture. Besides Specformer, we added the Polyformer in filter fitting experiments to further verify the learning capability of our filter.  Please refer to Table A3 in our response to Reviewer axD1's Weakness #4.
>
> >**Question #8**  'order-adaptive' and 'spectrum-adaptive' are vague. The term 'graph order' is used mistakenly.
>
> We would like to clarify the "order-adaptive" in this paper is actually 'filter order-adaptive' in the spectral domain, rather than the spatial-domain order or 'graph order'.  It refers to using the $k$-power of eigenvalues to approximate filters, analogous to polynomial filter approximation in the spectral domain. 'spectrum-adaptive' indicates that a full (or subset of) spectrum (eigenvalues) is used to approximate spectral filters, similar to the ability of the filter in Specformer. We will give a clear description to avoid the misunderstanding in our final version.
>
> >**Question #9**  Complexity claim
>
> The overall forward complexity of GrokFormer is $O(N(Nq+d^2)+KqM)$, where we have $q=N$ on small-scale datasets. It is lower than that of related methods, e.g., Specformer with a complexity of $O(N^2(q+d)+Nd^2)$.
>
> >**Question #10** Why not use full (regular) self-attention?
>
> We replaced linear self-attention with regular attention (G-Reg) and conducted experiments, which show that both achieve comparable performance (Table A2), as they both capture non-local spatial features. We chose linear attention for its better efficiency.
>
> ```
> Table A2. Accuracy comparison on attention type.
> ```
> | | Cora |Squirrel |Actor|
> | --- | --- | --- | --- |
> |G-Reg|89.43| 65.32|42.75|
> |GrokFormer |89.57| 65.12|42.98|

---

> > ### Comment · Reviewer_tPj2 · 2025-04-04
> >
> > Thanks for the response.
> >
> >
> > > Q1: (+)
> >
> > I agree that GrokFormer has an explict control on the filter-orders. However, it indicates the `filter-orders` are pre-determined by human rather than are not **adaptively learned** from data.
> > In contrast, Specformer is the one trying to learn filter-orders from data, even though it might not successfully do so.
> > Since you show that Graph Filter KAN (GFKAN) reaches better empirical performance,
> > I think this point is good except that you might need to rethink about the choice of word "adaptive", because the adaptive usually means "automatically learning from data" rather than "pre-determined as hyperparameter".
> >
> >
> > > Q2:
> >
> > Good
> >
> > > Q3
> >
> > Good
> >
> >
> > > Q4: "the spectral graph filter designed in GrokFormer is not based on the spatial-domain WL test. The GrokFormer is a spectral GT operating in the spectral domain while 1-WL or GD-WL algorithms are essentially spatial methods based on the graph structure"
> >
> > I disagree with this point. The concept of expressivity is not specific to the spatial domain.
> > It is a fundamental evaluation of the ability of graph models to distinguish different graph structures.
> > However, I agree that it might be difficult to directly compare against WL families.
> > But I still encourage the authors to figure this out. There are two potential routes to demonstrate the expressivity:
> > 1. (Theoretically) Wang \& Zhang (2022) provide a theoretical analysis on polynomial-based spectral GNNs on expressivity compared to 1-WL. This might provide some hints for you.
> > 2. (Empirically) Wang \& Zhang (2024) provide a benchmark to empirically evaluate the expressivity of graph models (ranging from 1-WL to 4-WL).
> >
> >
> > - Wang, Xiyuan, and Muhan Zhang. "How powerful are spectral graph neural networks." ICML 2022.
> > - Wang, Yanbo, and Muhan Zhang. "An Empirical Study of Realized GNN Expressiveness." ICML 2024
> >
> >
> > > Q5 and Q6
> >
> > #### First, Could you also provide the training accuracy on the experiments?
> > WikiCS, Pubmed, Squirrel and Actor are datasets of a single graph. The **small scale nature  of these datasets (in terms of the number of examples)** might be **insufficient to train graph Transformers**.
> > Given the fact that GRIT and GPS has much better performance on Peptides-func,
> > it is reasonable to question whether the worse performance is due to the worse capacity or overfitting issues.
> > (Note that the overfitting issue is very specific to the dataset rather than the model itself. It usually hints the stronger capacity of the model.)
> >
> > #### Second, considering the scale of the datasets utilize, please compare with other graph Transformers on ZINC, SP-CIFAR from [Ref1] to fully support your claim
> >
> > I ask for this since you propose a method as a graph Transformer. (Graph) Transformers are widely acknowledged to have stronger capacity, at the cost of a large amount of training data.
> >
> > > Q7
> > Honestly, PolyFormer and SpecFormer are arguably not regular Graph Transformers (which use regular attention mechanisms to process graphs directly).
> > PolyFormer computes the `attention` using $\text{tanh}(\mathbf{Q}\mathbf{K}^\intercal) \odot \mathbf{B}$ instead of Softmax. SpecFormer is *de facto* a spectral GNN with a Transformer encoder for eigenvalues, not a graph Transformers.
> >
> > However, it is fine at this stage. But it would be better for you to clearly distinguish the differences among them.
> >
> >
> > > Q8
> >
> > Good
> >
> > > Q9
> >
> >
> > Good
> >
> > > Q10
> >
> > Good
> >
> >
> >
> > > Final
> >
> > Regarding your response, I will raise the score to 2 temporarily.
> > If you could address Q5 and Q6, I will raise the score to 3. The Q4 might be difficult to address for now, but I encourage you to do so in the final version.

---

> > > ### Author Response · Authors · 2025-04-08
> > >
> > > Thank you very much for raising the score. We greatly appreciate your further comments and it is great to know that our response has addressed most of your questions. Please find our response to your remaining concerns below.
> > >
> > > >**Follow-up question with Question #1 #** About the adaptivity in 'filter-orders.'
> > >
> > > Thank you very much for pointing this out and the valuable suggestion. We would like to clarify that 'adaptivity' in the paper is to stress the adaptivity of combining varying orders of the filters. This adaptivity is achieved through the learnable parameters {$\alpha_k$} that enables an adaptive synergy of $K$ filter bases {$b_k(\lambda)$}, yielding the order-adaptive filter $h(\lambda) = \sum_{k=1}^{K}\alpha_kb_k(\lambda)$, where $k$ is an order of a filter.
> > >
> > > >**Follow-up question with Question #4 #**   The concept of expressivity is not specific to the spatial domain.
> > >
> > > We greatly appreciate the comments and the related literature you provided. We agree that expressivity is not limited to the spatial domain but is a fundamental measure of a graph model's capability. In this study, we have provided spectral analysis to show that our designed filter possesses universality and flexibility, and we will explore the two directions you suggested to have more in-depth discussion on the expressivity concept and the comparison with the WL family in our final version.
> > >
> > > >**Follow-up question with Question #5 #** The training accuracy on the experiments?
> > >
> > > Following your suggestion, we report the accuracy of each method on the training set in Table A4. As the results show,  both GraphGPS and GRIT achieve very high training accuracy on these small-scale node classification datasets, but their performance on the test sets is inconsistent with the training set. This can be attributed to the fact that these graph transformer-based methods dedicated to achieving higher expressive power are prone to overfit on these small datasets, leading to poor generalization on the test set. In contrast, GrokFormer focuses on designing an expressive spectral graph filter to capture diverse signals beyond low-frequency components in attention mechanisms, achieving better generalization ability on these datasets.
> > >
> > > ```
> > > Table A4. The training accuracy on four small-scale datasets.
> > > ```
> > > | | WikiCS|Pubmed|Squirrel|Actor|
> > > |--|--|--|--|--|
> > > |GraphGPS|0.9848|1.0|0.9405|0.9512|
> > > |GRIT|0.9814|1.0|0.9532|0.9781|
> > > |GrokFormer|0.9631|0.9502|0.9839|0.6275|
> > >
> > > >**Follow-up question with Question #6 #** Compare with other graph Transformers on ZINC, SP-CIFAR from [Ref1] to fully support your claim
> > >
> > > Following your suggestions,  to fully exploit the impact of the scale of the datasets and the capacity of the model, we further conducted the experiments on two additional datasets ZINC and SP-CIFAR, and the results are shown in Table. A5.
> > >
> > > ```
> > > Table A5. Comparison on two additional datasets (MAE on ZINC, classification accuracy on SP-CIFAR).
> > > ```
> > > | | ZINC |SP-CIFAR |
> > > | --- | --- | --- |
> > > | Graphormer |0.122|-|
> > > | GraphGPS   |0.070| 72.31 |
> > > | GRIT       |0.060| 75.67 |
> > > | GrokFormer |0.076| 73.75 |
> > >
> > > From the results, we observe that GrokFormer achieves better performance than Graphormer and performs comparably well to GraphGPS. GrokFormer slightly underperforms GRIT on ZINC while the performance margin on SP-CIFAR is relatively large. Considering all the empirical results, including both those on the suggested datasets and the small-scale datasets, it is clear that there is no method that consistently achieves the best performance across all datasets. GRIT is designed to improve GT's expressiveness in large datasets by incorporating graph inductive biases, so it can achieve more favorable performance on ZINC and SP-CIFAR but it has suboptimal performance on small-scale datasets due to over-fitting. GraphGPS achieves better performance on ZINC but shows suboptimal results on SP-CIFAR and the small-scale datasets compared to GrokFormer. As far GrokFomer, we aim to enhance the GT's capability to capture various frequency information on graphs by designing an expressive filter, which offers a relatively better balance between the generalization ability and the expressiveness, as demonstrated by the empirical results above. We hope this clarification helps address your concerns.
> > >
> > > >**Follow-up question with Question #7 #** PolyFormer and SpecFormer are arguably not regular Graph Transformers
> > >
> > > We really appreciate your insight into this problem. We will clarify this point in the final version of the paper.
> > >
> > > Your comments have been tremendously helpful in enhancing our work. Thank you very much again for your great efforts and time on our paper.

---

### Official Review · Reviewer_n2Go · 2025-03-09

**Overall Recommendation:** 4

**Summary:**

This paper proposes GrokFormer, a Transformer-based graph spectral model that introduces the expressive graph filter to the Transformer architecture, effectively capturing a wide range of frequency signals in an order- and spectrum-adaptive manner. Experiments on synthetic and real-world datasets show the effectiveness of the proposed method.

## update after rebuttal

The authors' responses have addressed my concerns. Hence, I'd like to keep my original score.

**Claims And Evidence:**

Yes. The authors conduct extensive experiments on synthetic and real-world datasets. And the experimental results demonstrate that GrokFormer can learn an expressive filter and outperform baselines on node-level and graph-level tasks.

**Essential References Not Discussed:**

N/A

**Experimental Designs Or Analyses:**

I think the designs of experiments are reasonable.

**Methods And Evaluation Criteria:**

Yes. The evaluation criteria in this paper follows the same setting in previous methods.

**Other Comments Or Suggestions:**

N/A

**Other Strengths And Weaknesses:**

Strengths:
1. The motivation is reasonable and the idea is easy-to-follow.

2. The paper develops a new graph Transformer that introduces a novel graph spectral filter learning approach in the Transformer architecture, which effectively capture a wide range of frequency signals in an order- and spectrum-adaptive manner.

3. Comprehensive experiments are convincing in my opinion.

4. The paper is clearly organized and well-written.

Weaknesses：

1.	The eigenvalue decomposition brings significant computational costs compared to polynomial paradigm. The authors are suggested to include the time consumption of preprocessing in Table 8, and demonstrate whether the proposed method achieves a trade-off between computational cost and expressiveness.

2.	Clarification from the authors about some guidelines to selecting order K could be helpful.

3.	The authors are encouraged to include a deeper discussion between GrokFormer, PolyFormer and Specformer,  as they are all spectral graph transformer methods.

**Questions For Authors:**

1. In Figures 4 and 7, as the order K increases, the performance on some datasets appears to change only marginally. Alternatively, clarification from the authors about some guidelines to selecting order K could be helpful.

2. How to understand the benifits of higher-order filter bases in Eq. (6) ?

**Relation To Broader Scientific Literature:**

N/A

**Theoretical Claims:**

The authors provide theoretical analysis and empirical evidence, showing the improved flexibility and expressivity of the GrokFormer filter over Specformer and Polynomial filters, which seems sound to me.

---

> ### Author Rebuttal · Authors · 2025-04-01
>
> We sincerely appreciate your constructive and positive comments on methodology and experiment designs. Please see our response to your comments one by one below.
>
> >**Weakness #1** The time consumption of preprocessing should be included in Table 8, and whether the proposed method achieves a trade-off between computational cost and expressiveness.
>
> Thanks for your valuable suggestion. Following your suggestions, we measured the computational overhead (including preprocessing) on the small-scale dataset Squirrel and the large-scale dataset Penn94, with the results summarized in Table A1 below.
>
> ```
> Table A1. The computational cost in terms of GPU memory required in MB, training runtime, and pre-processing time (s) on two datasets.
> ```
> | Dataset|Methods |Memory (MB)|Forward Times (s)|  Decomposition Time (s)|
> | --- | --- | --- | --- |---|
> |Squirrel|PolyFormer| 8678|42.38|/|
> ||Specformer| 1951|6.45|3.26|
> ||GrokFormer|1424|6.12|3.26|
> |Penn94|PolyFormer| 14113|121.78|33.14|
> ||Specformer| 5053|9.39|746.32|
> ||GrokFormer|4647|8.13|746.32|
>
> It is clear that our GrokFormer and Specformer are comparably fast, both of which requires much less computational cost than PolyFormer. Additionally, accuracy results of node classification on Squirrel show that spectral decomposition methods, such as GrokFormer and SpecFormer, achieve approximately a 20\% improvement over the polynomial filtering model PolyFormer. On the large-scale graph dataset Penn94, while preprocessing takes some time, the forward-pass complexity can be reduced using truncated spectral decomposition. Moreover, since spectral decomposition is computed only once and reused during training, its cost becomes a minor one compared to the forward-pass cost when training for many iterations.
>
> >**Weakness #2 and Question #1** Clarification from the authors about some guidelines to selecting order $K$ could be helpful.
>
> Please see our detailed response to ***Reviewer Cg7z’s Suggestion #1 and Question #3***, where we clarified the concern regarding the selection about filter order $K$.
>
> >**Weakness #3**  A deeper discussion between GrokFormer, PolyFormer, and Specformer should be included.
>
> Specformer designs a learnable filter by performing self-attention on each eigenvalue (spectrum), which is a spectrum-adaptive spectral method. PolyFormer designs an adaptive filter by performing self-attention on $K$ bases with fixed filter order. Different from these two methods, GrokFormer meticulously designs a universal order- and spectrum-adaptive filter within a Transformer architecture to capture diverse frequency components in graphs. We will add this comparison in the related work section.
>
> >**Question #2** How to understand the benefits of the higher-order filter base?
>
> Higher-order filter bases can result in the following two benefits: (1) enrich the spectral representation through additional frequency components, and (2) strengthen the approximation power. These advantages facilitate more accurate signal distribution modeling in diverse graphs of various complexities. Our method leverages these advantages through an adaptive order selection mechanism that dynamically optimizes filter orders for datasets with varying properties, achieving large performance improvement over existing SOTA filters, e.g., achieving 21\% improvement over the baseline PolyFormer with fixed-order filter bases on the Squirrel dataset.

---

> > ### Comment · Reviewer_n2Go · 2025-04-02
> >
> > Thank you for your responses. After reading comments of other reviewers as well as the corresponding responses, I’d like to keep my score. I hope you can add the above results and discussions in the revised version.

---

> > > ### Author Response · Authors · 2025-04-02
> > >
> > > We truly appreciate your timely and valuable feedback on our paper. Your thoughtful comments are extremely helpful for enhancing our work. We will carefully prepare the final version of our paper based on the experimental results and discussions provided in the rebuttal.

---

### Official Review · Reviewer_Cg7z · 2025-03-13

**Overall Recommendation:** 3

**Summary:**

The paper introduces GrokFormer, a novel Graph Transformer (GT) model that addresses limitations in existing graph learning methods, particularly in capturing diverse frequency signals in graph data. GrokFormer incorporates a Graph Fourier Kolmogorov-Arnold Network (KAN) to design spectral filters that are both spectrum- and order-adaptive. Unlike prior models with fixed or limited adaptability, GrokFormer uses Fourier series-based learnable activation functions to flexibly model graph Laplacian spectra across multiple orders. The model demonstrates superior expressiveness and adaptability compared to state-of-the-art (SOTA) Graph Neural Networks (GNNs) and GTs. Extensive experiments on node classification and graph classification tasks across various datasets validate its effectiveness, showing consistent improvements over baselines.

**Claims And Evidence:**

The paper makes several claims:

Expressiveness of GrokFormer Filters: The authors claim that GrokFormer filters are more expressive than existing spectral methods due to their spectrum- and order-adaptive capabilities. This is supported by theoretical proofs (e.g., Propositions 4.1–4.4) and empirical results on synthetic datasets, where GrokFormer outperforms others in fitting complex filters like comb and low-comb patterns.

Superiority in Real-World Tasks: GrokFormer achieves state-of-the-art performance on 10 node classification datasets and 5 graph classification datasets, outperforming both GNNs and GTs. For example, it achieves the highest accuracy on heterophilic datasets like Squirrel, where complex frequency responses are crucial.

Efficiency: The model is computationally efficient compared to alternatives like Specformer, as demonstrated by empirical time and memory complexity analyses.

**Essential References Not Discussed:**

The paper appears comprehensive in its citations but could benefit from discussing recent advancements in scalable spectral methods or alternative transformer architectures that address similar challenges.

**Experimental Designs Or Analyses:**

The experimental design is sound:

Experiments cover diverse datasets with varying scales, homophily levels, and graph properties.

Ablation studies isolate the contributions of key components (e.g., self-attention vs. Graph Fourier KAN).

Scalability tests demonstrate the model's efficiency on large graphs.

However, further exploration of hyperparameter sensitivity could enhance the robustness of conclusions.

**Methods And Evaluation Criteria:**

The methods proposed align well with the problem of capturing diverse frequency signals in graphs:

The use of Fourier series for filter modeling ensures flexibility across spectral frequencies.

The evaluation includes a wide range of datasets with varying properties (homophilic and heterophilic), ensuring robustness.

Baselines include both spatial-based GNNs (e.g., GCN, GAT) and spectral-based GTs (e.g., Specformer), providing a comprehensive comparison.

**Other Comments Or Suggestions:**

Clarify how hyperparameters like K (order) affect performance across different datasets.

Include runtime comparisons on larger-scale graphs beyond Penn94 for broader scalability insights.

**Other Strengths And Weaknesses:**

Strengths:

Original combination of Fourier series modeling with Transformer architecture.

Comprehensive theoretical analysis supporting empirical findings.

Strong performance across diverse datasets.

Weaknesses:

Limited discussion on potential limitations or failure cases.

Scalability could be further improved by avoiding spectral decomposition altogether.

**Questions For Authors:**

How does GrokFormer handle dynamic graphs where the topology changes over time?

Can the model's efficiency be improved further by avoiding spectral decomposition entirely?

How sensitive is the performance to hyperparameters like K or the number of Fourier terms (M)?

**Relation To Broader Scientific Literature:**

The paper builds on prior work in GNNs (e.g., ChebyNet, GPRGNN) and GTs (e.g., Specformer). It advances the field by addressing limitations in spectrum adaptability (Specformer) and order adaptability (polynomial GNNs). The use of Fourier series aligns with established theories in approximation and signal processing.

**Theoretical Claims:**

The theoretical claims are supported by detailed proofs:

Proposition 4.1 shows the learnability of GrokFormer filters in both polynomial order and spectrum.

Proposition 4.2 demonstrates that polynomial filters used in existing methods are special cases of GrokFormer filters.

Proposition 4.3 establishes that Specformer is a simplified variant of GrokFormer.

Proposition 4.4 proves the universal approximation capability of GrokFormer filters using Fourier series.

---

> ### Author Rebuttal · Authors · 2025-04-01
>
> We sincerely appreciate your constructive and positive comments on methodology design and theoretical analysis. Please see our response to your comments one by one below.
>
> > **Weakness #1** Limited discussion on potential limitations or failure cases.
>
> Thank you very much for the suggestion and question. We will provide a more comprehensive discussion of GrokFormer's potential limitations. One limitation is that when learning optimal spectral filters that requires the complete spectrum, the full spectral decomposition may lead to a huge computational cost on very large datasets, but this limitation do not affect the detection effectiveness of GrokFormer on a variety of real-world graph datasets. Please also find another limitation in our response to Reviewer axD1's Weakness #1 and #2, where we discuss that our spectral graph filter may miss some spatial structure information in some datasets.
>
> > **Weakness #2, Question #2, and Suggestion #2** Can the model's efficiency be improved further by avoiding spectral decomposition entirely? Include runtime comparisons on larger-scale graphs beyond Penn94 for broader scalability insights.
>
> Thank you for your insightful suggestion. Avoiding spectral decomposition entirely can further enhance the model's efficiency. To reduce computational costs, we can employ truncated spectral decomposition to compute only the top-$p$ eigenvalues in large-scale graphs.  As shown in Table A1, we provide runtime comparisons on the arXiv dataset, which contains 169,343 nodes (larger than Penn94). By considering only the smallest 5,000 eigenvalues, our model remains computationally efficient even on large-scale datasets.
>
> ```
> Table A1. The training cost in terms of GPU memory (MB) and running time (in seconds) on arXiv.
> ```
> | |Memory (MB)|Times (s)|
> | --- | --- |---|
> |PolyFormer|4019| 296.53 |
> |Specformer|5565 |6.58|
> |GrokFormer|5491|6.37|
>
> > **Suggestion #1 and Questions #3** Clarify how hyperparameters like $K$ (order) affect performance across different datasets. How sensitive is the performance to hyperparameters like $K$ or the number of Fourier terms ($M$)?
>
> Thank you for the valuable comments. In our model, both filter order $K \in [1,6]$ and Fourier terms $M \in [32,64]$ can empower the filter to include richer frequency components and enhance the fitting capability of filter.  From the datasets description in Table 6 and the experimental results regarding the filter order $K$ in Figures 3 and 7, we can observe that (1) A smaller $K$ value is suitable for homophilic datasets and small-scale heterophilic datasets with sparse edges; (2) A larger $K$ value is suitable for large-scale heterophilic datasets and heterophilic datasets with dense edges. Similar observations can be obtained for Fourier terms ($M$), too.
>
> > **Questions #1** How does GrokFormer handle dynamic graphs where the topology changes over time?
>
> Like most competing methods, GrokFormer is designed for static graphs. Dynamic graphs, where the topology evolves over time, are not applicable in the current setting of these methods. We plan to extend GrokFormer to dynamic graphs in our future work.

---

### Official Review · Reviewer_axD1 · 2025-03-25

**Overall Recommendation:** 4

**Summary:**

This paper proposes GrokFormer, a novel graph transformer (GT), with superior capability in modelling complex spectral filters. The filter design is both order and spectrum adaptive and is implemented using a specific instantiation of Kolmogorov-Arnold Network in the spectral domain. Results on several node and graph level tasks demonstrate its effectiveness. The visualizations show the expressive modelling of the designed filters. Specifically, comparison with Specformer for modelling complex, comb-like frequency patterns shows the superiority of this novel parameterization of the filter.

**Update**: I have carefully read the other reviews and the authors' rebuttals. Overall, the new results presented during the rebuttal period improve the paper greatly. Please revise the paper by including these new results and discussions if accepted. I am happy to raise the score to 4 to indicate an accept recommendation.

**Claims And Evidence:**

Theoretical claims: (proofs are in Appendix C)

1) $h(\lambda)$ is learnable in both polynomial order and graph spectrum.

2) Existing spectral GNNs' filters can be represented by this approach.

3) The graph filter in Specformer is a simplified variant of proposed filter

4) $h(\lambda)$ can approximate any continuous function and constructs a permutation-equivariant spectral graph convolution (similar proof as in Specformer).

Empirical claims:

1) Better benchmarking results on both real and synthetic data: Table 2-4

2) More flexible modelling of the frequency spectrum: Figures 1, 5

**Essential References Not Discussed:**

N/A

**Experimental Designs Or Analyses:**

Please see the comments on the experimental evaluation below.

**Methods And Evaluation Criteria:**

The numerical evaluation is  generally comprehensive (specific questions/comments in later sections). In addition to standard transductive node classification benchmarks (both homophilic and heterophilic graphs), several small-scale TU benchmarks for graph classification is also considered. Experiment on the synthetic dataset in Section 5.3 (also studied in Specformer) also shows the impressive modelling capability of GrocFormer.

**Other Comments Or Suggestions:**

N/A

**Other Strengths And Weaknesses:**

Strengths:

1) The paper is well-written, easy to read. The visualizations on both synthetic and real data are intriguing.

2) The experiments are thorough and substantiates the main idea clearly for the chosen benchmarks.

Weaknesses:

1) There is no experiments on the popular Benchmarking GNN datasets [1], which are the de-facto benchmarks in recent graph transformers, e.g., GRIT [2]. It is well-known that those benchmarks are more difficult and requires higher expressivity compared to the datasets considered in this paper. For example, the node classification is experimentally studied entirely in the transductive setting in this work, whereas [1] contains inductive node classification benchmarks.

2) Similarly, it would strengthen the paper if the authors include some results on the long range graph benchmark (LRGB) [3] datasets.

3) It would be helpful to characterize the theoretical expressivity of GrocFormer in terms of GD-WL [4], and compare and contrast it with existing GTs, e.g. GRIT [2].

4) While the authors cite (Xu et al., 2024) for motivating the specific design of the filter in eq. 5, it would be more convincing if some numerical results are presented (even on the synthetic dataset) to showcase why this specific parameterization is chosen. What other alternatives were considered? How were those results compared to this version of the filter?


References:
[1] Dwivedi, V. P., Joshi, C. K., Laurent, T., Bengio, Y., and Bresson, X. Benchmarking Graph Neural Networks. J. Mach. Learn. Res., December 2022a.

[2]  Ma, L., Lin, C., Lim, D., Romero-Soriano, A., K. Dokania, P., Coates, M., H.S. Torr, P., and Lim, S.-N. Graph Inductive Biases in Transformers without Message Passing. In Proc. Int. Conf. Mach. Learn., 2023.

[3] Dwivedi, V. P., Rampa´sek, L., Galkin, M., Parviz, A., Wolf, ˇG., Luu, A. T., and Beaini, D. Long Range Graph Benchmark. In Adv. Neural Inf. Process. Syst., December 2022b.

[4] Zhang, B., Luo, S., Wang, L., and He, D. Rethinking the Expressive Power of GNNs via Graph Biconnectivity. In Proc. Int. Conf. Learn. Represent., 2023.

**Questions For Authors:**

Please address the questions/comments in the preceding sections.

**Relation To Broader Scientific Literature:**

This work improves upon Specformer by proposing a more flexible KAN-based parameterization of the filter in the spectral domain, which has superior capability of modelling complicated spectral characteristics.

**Theoretical Claims:**

The proofs are written clearly, relatively simple to follow and are correct to the best of of my knowledge.

---

> ### Author Rebuttal · Authors · 2025-04-01
>
> We sincerely appreciate your constructive and positive comments on the presentation, visualization, and experiment design. Please see our response to your comments one by one below.
>
> >**Weaknesses #1 and #2**  Lack of a few challenging benchmark GNN datasets (datasets in [Ref1] and LRGB datasets [Ref2])  and two recent baseline methods (GraphGRS [Ref3] and GRIT [Ref4]).
>
> Following your suggestion, we conducted the experiments on the two representative datasets, WikiCS and Peptides-func, obtained from [Ref1] and [Ref2] respectively, to further demonstrate the expressiveness of GrokFormer.  Additionally, we also added the two new expressive GTs, GraphGPS and GRIT, as our competing methods. The experimental results are shown in Tables A1 and A2.
>
> ```
> Table A1. Classification accuracy (in percent) on the WikiCS and Test AP on Peptides-func.
> ```
> | |WikiCS|Peptides-func|
> | --- | --- |---|
> |Specformer|82.37|0.5771|
> |Polyformer|80.19|0.5319|
> |GraphGPS|78.66|0.6257|
> |GRIT|77.21|0.6458|
> |GrokFormer|83.11|0.5987|
>
> ```
> Table A2. Classification accuracy comparison of GRIT, GraphGPS, and GrokFormer.
> ```
> ||Pubmed|Squirrel|Actor|
> | --- | --- |---|---|
> |GraphGPS|86.94|35.58|36.01|
> |GRIT|87.62|37.68|37.51|
> |GrokFormer|91.39|65.12|42.98|
>
> The experimental results show that GrokFormer consistently outperforms all competing methods, including the newly added expressive GTs, in Table A2 and on WikiCS of Table A1. This is primarily because expressive GTs like GraphGPS and GRIT struggle to capture the underlying label patterns due to the Transformer's low-pass nature, resulting in suboptimal performance. Note that the spectral-based methods slightly underperform the expressive GTs that are specifically designed to capture the major spatial structure information in Peptides-func, which is not modeled by these general spectral methods.  Thus, we believe that this suboptimal performance does not diminish our contribution in the proposed expressive spectral graph filters, considering its ability in enhancing the Transformer's capability to capture frequency signals beyond low-pass on graphs.
>
> - [Ref1] Benchmarking Graph Neural Networks. JMLR, 2022.
>
> - [Ref2] Long Range Graph Benchmark.  NeurIPS, 2022.
>
> - [Ref3] Recipe for a general, powerful, scalable graph transformer. NeurIPS, 2022.
>
> - [Ref4] Graph Inductive Biases in Transformers without Message Passing. ICML, 2023.
>
> >**Weakness #3**  The theoretical expressivity discussion in terms of WL, and compare and contrast it with existing GTs, e.g.  GRIT.
>
> We would like to clarify that the spectral graph filter designed in GrokFormer is not based on the spatial-domain WL test. The GrokFormer is a spectral GT operating in the spectral domain while 1-WL or GD-WL algorithms are essentially spatial methods based on the graph structure. Therefore, GRIT exhibits better localization spatially, while our GrokFormer achieves better localization on frequencies. Although our work lacks a formal theoretical analysis in the spatial domain to discuss strong expressivity like the WL test, we provide theoretical analysis in the spectral domain to justify its advantages over existing advanced filters and show that our filter possesses better universality and flexibility. Moreover, the filter fitting experiment on synthetic datasets and comprehensive empirical comparison on real-world datasets demonstrate the stronger expressivity of our GrokFormer filter.
>
> >**Weakness #4** Lack of numerical results for the filter design in Eq. (5). What other alternatives were considered? How were those results compared to this version of the filter?
>
> We would like to clarify that we have pre-defined six different filters on the synthetic dataset in Section 5.3.1 to verify the fitting capability of the filter design in Eq.(5) compared to alternative methods. Here we present partial results in Table A3 as follows,
>
> ```
> Table A3. Node regression results on two difficult synthetic datasets.
> ```
> | |High-pass|Comb|
> | --- | --- |---|
> |GPRGNN|0.1046(.9985)|4.9416(.9283) |
> |Specformer|0.0029(.9999) |0.0083(.9998)|
> |PolyFormer|0.0482(0.9993)|0.2868(0.9945)|
> |GrokFormer|0.0012(.9999)|0.0021(.9999)|
>
> Note that we use two metrics to evaluate each method: sum of squared error ($R^2$ score). Lower SSE (higher $R^2$) indicates better performance. The effectiveness of joint filter order-adaptive and spectrum-adaptive properties in Eq.(5) is empirically confirmed through its much better performance on such difficult synthetic datasets. Meanwhile, two simplified alternative cases (a) spectrum-only adaptation without high-order filter terms and (b) filter order-only adaptation are considered in the comparison above.  Per Propositions 4.2 and 4.3, the filter variant in case (a) can be regarded as the Specformer filter, while the variant in case (b) corresponds to the GPRGNN filter. The results show that heir fitting ability of the GrokFormer filter is much stronger than the two alternatives.

---

> > ### Comment · Reviewer_axD1 · 2025-04-03
> >
> > I thank the authors for their response, it addresses most of my comments.
> >
> > I have only one remaining question.
> >
> > **numerical results for the filter design in Eq. (5). What other alternatives were considered? How were those results compared to this version of the filter?**   I have read and understood the effectiveness of proposed method in comparison to the baselines for  the synthetic dataset  experiment in Section 5.3.1. My question was specifically about the chosen parameterization in Eq. 5? What is the specific motivation of using sine and cosine instead of spline? Were any other alternatives considered? If yes, what's the empirical comparison with that?

---

> > > ### Author Response · Authors · 2025-04-04
> > >
> > > We're very pleased to know that our response has addressed most of your concerns. We really appreciate your further comments, and please find our response to your follow-up question as follows.
> > >
> > > >**Follow-up question with Question #1 #**  What is the specific motivation of using sine and cosine instead of spline in the chosen parameterization in Eq. (5)?
> > >
> > > The motivation for choosing sine and cosine functions is that they can be viewed as components of a Fourier series, allowing the filter to approximate any continuous function over the eigenvalue range [0,2]; furthermore, this setting helps retain orthogonality in Fourier series, supporting the learning of diverse complementary frequency components within a specific interval.
> > >
> > > >**Follow-up question with Question #2 #** Any other alternatives considered and the empirical comparison?
> > >
> > > As commonly used for filter functions, exponential functions (e.g.,  $e^{-x}$) can be considered as an alternative to sine and cosine functions,  such as GraphHeat [Ref1]. Since $e^{-x}$ mainly captures low-frequency component, to better captures diverse frequency components (e.g., low, high, and band-pass), we combine it with a trigonometric function (e.g., cosine) to derive an alternative filter function $h(\lambda)$=$\sum_{k=1}^{K}\sum_{m=0}^{M}$($a_{km}$*cos($m\lambda^k$)+$b_{km}$*$e^{-\lambda^k}$).
> > > We refer to this variant as GrokFormer$^*$ below. We compared GrokFormer$^*$ with GrokFormer on synthetic datasets in Table A1 and real-world datasets in Table A2 to evaluate its learning capability.
> > >
> > > ```
> > > Table A1. Node regression results on synthetic datasets.
> > > ```
> > > ||Low-pass|High-pass|Band-pass|Band-rejection|Comb|Low-comb|
> > > |--- | --- |---|--- | --- |---|---|
> > > |GrokFormer$^{*}$|0.0085(.9999)|0.0073(.9999)|0.0102(.9999)|0.0166(.9999)|0.0205(.9998)|0.0435(.9998)|
> > > |GrokFormer|**0.0011(.9999)**|**0.0012(.9999)**|**0.0004(.9999)**|**0.0024(.9999)**|**0.0021(.9999)**|**0.0029(.9999)**|
> > >
> > >
> > > ```
> > > Table A2. Node classification results on the real-world datasets.
> > > ```
> > > ||Cora|Citeseer|Pubmed|Squirrel|Chameleon|Actor|WikiCS|
> > > |--- | --- |---|--- | --- |---|---|---|
> > > |GrokFormer$^{*}$|88.69|80.71|90.29|63.98|72.16|41.67|82.54|
> > > |GrokFormer|**89.57**|**81.92**|**91.39**|**65.12**|**75.58**|**42.98**|**83.11**|
> > >
> > > Based on the results in Tables A1 and A2, we can observe that GrokFormer$^*$ has consistently less effective filter-fitting capability and classification performance than GrokFormer on synthetic and real-world datasets, respectively. In particular, GrokFormer tends to obtain a larger superiority over GrokFormer$^*$ on the heterophilic datasets. This may be due to that, as discussed above, the alternative filter function in GrokFormer$^{*}$ fails to achieve certain advantageous properties of the filter function in Eq. (5), such as orthogonality and its ability of learning more diverse frequency components.
> > >
> > >
> > > We hope the above reply helps address this follow-up question. We will discuss and clarify this point in our final version. We're more than happy to engage in more discussion with you to address any further concerns you may have. Thank you very much for helping enhance our paper again!
> > >
> > > - [Ref1] Xu, B., Shen, H., et al. Graph Convolutional Networks using Heat Kernel for Semi-supervised Learning. IJCAI, 2020.

---

### Decision · Program_Chairs · 2025-05-01

**Decision:**

Accept (poster)

**Comment:**

This paper presents GrokFormer, a novel Graph Transformer that leverages adaptive spectral filters via Fourier series modeling to address limitations in capturing diverse frequency signals in graph data. The reviewers highlight its theoretical contributions, including proofs of expressiveness and flexibility, as well as strong empirical performance across 10 node and 5 graph classification datasets. While some reviewers initially raised concerns about comparisons with expressive graph Transformers (e.g., GRIT, GraphGPS) and the clarity of "adaptivity," the authors' rebuttal addressed these issues by adding experiments on challenging benchmarks (e.g., ZINC, SP-CIFAR) and clarifying terminology. The method's ability to balance generalization and expressiveness, particularly on heterophilic graphs, was noted as a strength. Although scalability and dynamic graph handling remain future work, the consensus leans toward acceptance due to the paper's solid theoretical grounding, comprehensive evaluations, and rebuttal improvements. Given its contributions to spectral graph learning and Transformer architectures, we recommend acceptance with the expectation that the authors incorporate the suggested revisions (e.g., expanded discussions on expressivity and dataset comparisons) in the final version.